# Gα$_i$ is required for carvedilol-induced β$_1$ adrenergic receptor β-arrestin biased signaling

Jialu Wang[1,2], Kenji Hanada[2], Dean P. Staus[2], Michael A. Makara[2], Giri Raj Dahal[2], Qiang Chen[2], Andrea Ahles[3], Stefan Engelhardt [3,4] & Howard A. Rockman[1,2,5]

The β$_1$ adrenergic receptor (β$_1$AR) is recognized as a classical Gα$_s$-coupled receptor. Agonist binding not only initiates G protein-mediated signaling but also signaling through the multifunctional adapter protein β-arrestin. Some βAR ligands, such as carvedilol, stimulate βAR signaling preferentially through β-arrestin, a concept known as β-arrestin-biased agonism. Here, we identify a signaling mechanism, unlike that previously known for any Gα$_s$-coupled receptor, whereby carvedilol induces the transition of the β$_1$AR from a classical Gα$_s$-coupled receptor to a Gα$_i$-coupled receptor stabilizing a distinct receptor conformation to initiate β-arrestin-mediated signaling. Recruitment of Gα$_i$ is not induced by any other βAR ligand screened, nor is it required for β-arrestin-bias activated by the β$_2$AR subtype of the βAR family. Our findings demonstrate a previously unrecognized role for Gα$_i$ in β$_1$AR signaling and suggest that the concept of β-arrestin-bias may need to be refined to incorporate the selective bias of receptors towards distinct G protein subtypes.

[1] Department of Cell Biology, Duke University Medical Center, Durham, NC 27710, USA. [2] Department of Medicine, Duke University Medical Center, Durham, NC 27710, USA. [3] Institute of Pharmacology and Toxicology, Technical University of Munich, Munich 80802, Germany. [4] German Center for Cardiovascular Research (DZHK), Partner Site Munich Heart Alliance, Munich 80802, Germany. [5] Departments of Molecular Genetics and Microbiology, Duke University Medical Center, Durham, NC 27710, USA. Correspondence and requests for materials should be addressed to H.A.R. (email: h.rockman@duke.edu)

G protein-coupled receptors (GPCRs) represent the largest and the most versatile family of cell surface receptors[1]. Members of this receptor family translate diverse extracellular cues to intracellular responses, and are commonly targeted for medicinal therapeutics[2, 3]. One of the most commonly used therapeutic agents in medicine are ligands that target β adrenergic receptors (βARs) because they regulate many important physiological processes involved in the regulation of cardiovascular and pulmonary function[4].

GPCRs selectively couple to different heterotrimeric G protein complexes (Gαβγ) that are classified into four families based on their α-subunits: $G\alpha_{stimulatory}$ ($G\alpha_s$), $G\alpha_{inhibitory/olfactory}$ ($G\alpha_{i/o}$), $G\alpha_{q/11}$, and $G\alpha_{12/13}$[5]. Among the different G protein subtypes, βARs primarily transmit signals through $G\alpha_s$[6]. In the classical paradigm of βAR signaling, receptors exist in two distinct conformational states: active or inactive. Agonist binding stabilizes an active βAR conformation that promotes coupling with heterotrimeric G proteins, triggering guanine nucleotide exchange of $G\alpha_s$ and its dissociation from the Gβγ subunits, leading to the activation of adenylyl cyclase and triggering second messenger cyclic AMP signaling[7, 8]. Subsequent to agonist binding, activated βARs are phosphorylated by G protein-coupled receptor kinases (GRKs) leading to recruitment of the multifunctional β-arrestins (β-arrestin1 and β-arrestin2) and inhibition of further G protein coupling, a process termed desensitization[8]. It is now appreciated that β-arrestins also act as signal transducers in their own right[7] to stimulate a distinct array of signaling and cellular responses, such as transactivation of the epidermal growth factor receptor (EGFR)[9, 10], induction of extracellular signal-regulated kinase (ERK)[10–13], and activation of $Ca^{2+}$/calmodulin kinase II (CaMKII)[14]. Current data suggest a much greater complexity of GPCR signaling than the two-state (active or inactive) model whereby multiple receptor conformations can exist, each with a different affinity for its transducer, resulting in the activation of distinct cellular signaling pathways[15–17]. Whereas balanced ligands, such as isoproterenol, stabilize βAR conformations signal with equal efficacy through G proteins and β-arrestins, some ligands stabilize conformations that selectively recruit only one of the transducers to stimulate a specific subset of cellular signals, a process termed "biased agonism"[18, 19]. As biased ligands may be capable of selectively activating beneficial signaling while simultaneously blocking untoward receptor activated pathways[20], understanding mechanisms of biased agonism can have important implications for drug discovery targeting GPCRs.

The β-blocker carvedilol is a β-arrestin-biased βAR ligand that preferentially activates β-arrestin-mediated pathways while having inverse agonism towards $G\alpha_s$ signaling[7, 10, 19, 21]. To date, the prevailing mechanistic concept of β-arrestin-bias for the $G\alpha_s$-coupled $\beta_1AR$ is ligand-stimulated activation of β-arrestin in the absence of G protein coupling. However, recently it has been demonstrated for the angiotensin II type 1 receptor that the weak β-arrestin-biased agonist, [$^1$Sar$^4$Ile$^8$Ile]-angiotensin II, is capable of activating both $G\alpha_q$ and $G\alpha_i$[22], indicating a possible role of G proteins in β-arrestin-mediated signaling. Moreover, recent biophysical work suggests that both G protein and β-arrestin can simultaneously interact with an activated GPCR to form super complexes[23], raising the possibility that association of β-arrestin with the receptor may not preclude interaction with a G protein. Here, we set out to test whether G protein coupling is a critical component of β-arrestin-biased βAR signaling. Our findings show that carvedilol, unique among other βAR agonists or antagonists tested, selectively promotes the recruitment of $G\alpha_i$ to $\beta_1ARs$ to initiate β-arrestin-biased signaling. These data underscore the complexity of β-arrestin-biased agonism and have important implications for identifying new therapeutic agents to selectively target β-arrestin-biased signaling.

## Results

**$G\alpha_i$ is required for carvedilol-induced $\beta_1AR$-mediated ERK.** Previous studies have demonstrated that carvedilol induces βAR-mediated ERK phosphorylation in a $G\alpha_s$-independent, β-arrestin-dependent manner[10, 21]. To determine whether $G\alpha_i$ is required for carvedilol-stimulated βAR signaling, we tested the effect of the $G\alpha_i$ inhibitor pertussis toxin (PTX) on carvedilol-stimulated ERK phosphorylation in HEK293 cells stably expressing FLAG-tagged $\beta_1AR$ or $\beta_2AR$. PTX catalyzes the ADP-ribosylation of $G\alpha_i$ and prevents $G\alpha_i$ coupling to ligand bound receptors. In $\beta_1AR$ stable cells, carvedilol dose dependently increased ERK phosphorylation, which was significantly diminished by pretreatment with the $G\alpha_i$ inhibitor PTX (Fig. 1a, Supplementary Fig. 1a). In contrast, PTX had no effect on the carvedilol-induced $\beta_2AR$-mediated ERK phosphorylation (Fig. 1a, Supplementary Fig. 1a). These observations suggest that $G\alpha_i$ is needed for carvedilol-induced $\beta_1AR$, but not $\beta_2AR$ signaling.

To further delineate the role of $G\alpha_i$ in carvedilol-induced βAR signaling, we measured the level of ERK activation in $\beta_1AR$ or $\beta_2AR$ stable cells after removing $G\alpha_i$ using CRISPR/Cas9 gene editing. All three subtypes of $G\alpha_i$ ($G\alpha_{i1}$, $G\alpha_{i2}$, and $G\alpha_{i3}$) were depleted with their specifically targeted guide RNAs (Supplementary Fig. 1b). $G\alpha_i$ depletion markedly blocked carvedilol-induced ERK phosphorylation in $\beta_1AR$ stable cells, while it had no effect in $\beta_2AR$ stable cells (Fig. 1b). The absence of $G\alpha_i$ was considerably more robust in abrogating carvedilol-stimulated ERK phosphorylation compared to that observed with PTX treatment (Fig. 1a).

We then determined if a similar signaling mechanism is involved in heart tissue by measuring ERK phosphorylation in Langendorff perfused mouse hearts following carvedilol stimulation. To study the specific effect of carvedilol on the $\beta_1AR$, we used previously generated $\beta_2AR$ knockout mice[24]. Carvedilol perfusion robustly stimulated ERK phosphorylation in hearts of $\beta_2AR$ knockout mice, which was entirely abrogated in hearts of PTX-pretreated mice (Fig. 1c). In contrast, in $\beta_1AR$ knockout mice[25] while carvedilol robustly induced ERK phosphorylation by activating the $\beta_2AR$, PTX pretreatment was unable to block ERK activation (Fig. 1c). These data are consistent with our in vitro data and indicate a previously unrecognized, βAR subtype specific, requirement for $G\alpha_i$ in carvedilol-induced $\beta_1AR$ signaling.

**Carvedilol-induced $\beta_1AR$ conformational change requires $G\alpha_i$.** Different ligands for the same receptor stabilize unique conformational states promoting coupling to selective signal transducers and activation of distinct downstream signaling pathways[7, 20]. Since we showed that $G\alpha_i$ is required for carvedilol-induced $\beta_1AR$ signaling, we tested whether it allosterically stabilizes a unique carvedilol-bound $\beta_1AR$ conformation. We utilized a fluorescence resonance energy transfer (FRET)-based $\beta_1AR$ conformational sensor in which Cerulean (Cer) and YFP are inserted in the C-terminus and third intracellular loop of the receptor, respectively (Fig. 2a)[26]. Agonist-induced $\beta_1AR$ activation is represented by the loss of FRET, i.e., decrease of YFP/Cer ratio[26]. To test whether $G\alpha_i$ stabilizes a carvedilol-induced $\beta_1AR$ conformation, HEK293 cells stably expressing the $\beta_1AR$ FRET sensor were pretreated with vehicle or PTX, then stimulated with the balanced agonist isoproterenol or the β-arrestin-biased agonist carvedilol while monitoring the FRET ratio in real time. We found that compared to isoproterenol which caused a decrease in the FRET ratio, carvedilol induced a directional opposite response to the FRET signal, whereas the $\beta_1AR$ antagonist metoprolol showed no effect (Fig. 2b). Importantly, pretreatment with PTX significantly diminished the carvedilol-induced FRET ratio without any effect on the isoproterenol stimulated FRET-based receptor biosensor (Fig. 2c). Lastly, PTX alone did not affect the

FRET ratio (Supplementary Fig. 2). These data demonstrate the $\beta_1AR$ adopts a distinct conformational state when bound to iso-proterenol compared to carvedilol and that $G\alpha_i$ is needed to stabilize the carvedilol-bound $\beta_1AR$ conformation.

**Carvedilol selectively promotes $G\alpha_i$ recruitment to $\beta_1ARs$.** To determine the mechanism of how $G\alpha_i$ is involved in carvedilol-induced $\beta_1AR$ signaling, we measured ligand-promoted $G\alpha_i$ recruitment to $\beta ARs$ with an in situ proximity ligation assay

(PLA), a confocal-microscopy based assay that allows direct visualization and quantification of protein–protein interactions. Using HEK293 cells stably expressing $\beta_1ARs$, we show an over twofold increase in the PLA signal after carvedilol treatment, indicating recruitment of $G\alpha_i$ to the $\beta_1AR$ (Fig. 3a). In contrast, carvedilol had no effect on the recruitment of $G\alpha_i$ to $\beta_2ARs$, but $G\alpha_i$ was robustly recruited by isoproterenol consistent with the known process of G protein switching for $\beta_2ARs$[27] (Fig. 3a). Importantly, pretreatment with the $\beta AR$ antagonist propranolol blocked the carvedilol-induced $G\alpha_i$ recruitment to $\beta_1ARs$

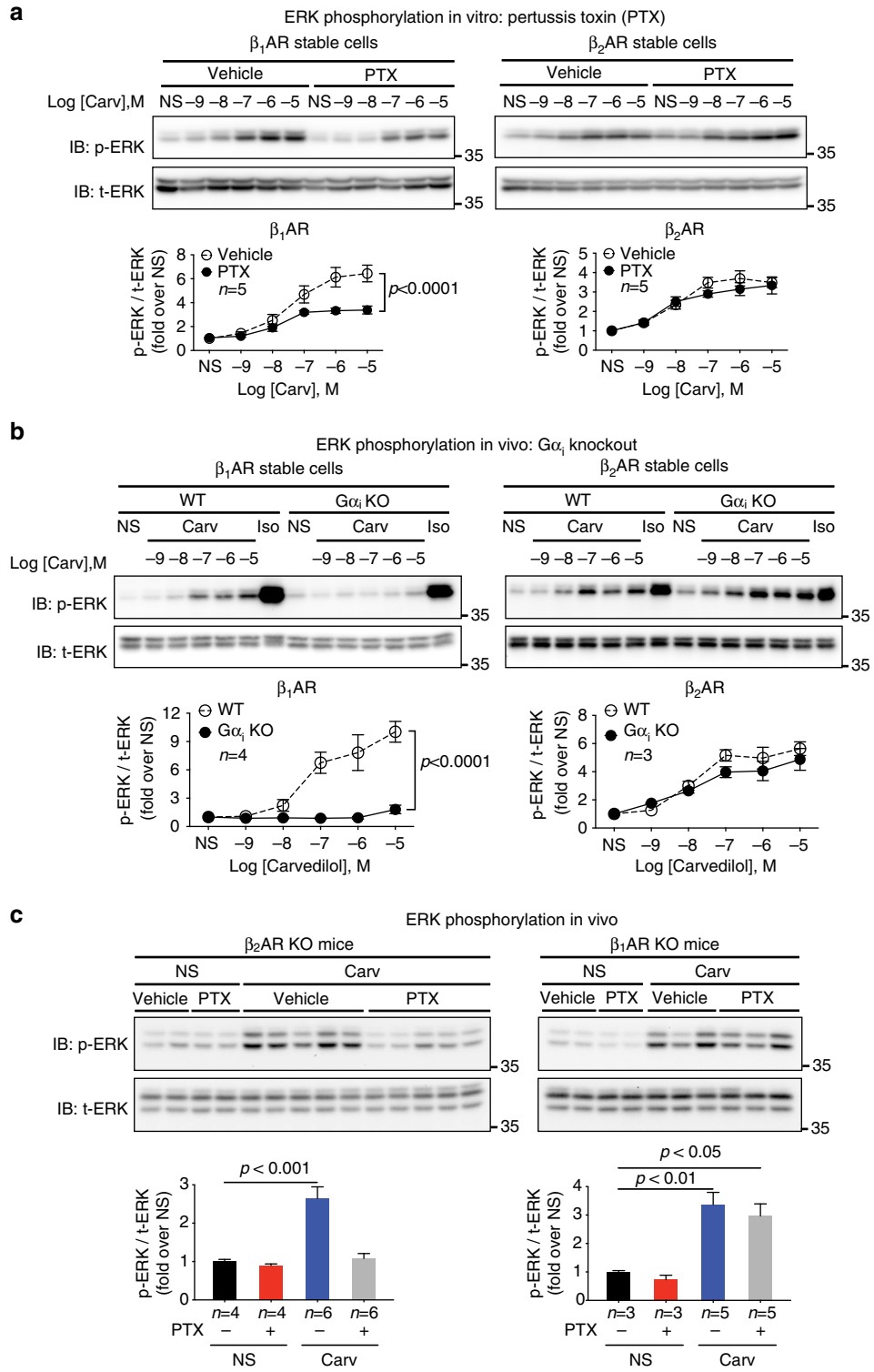

(Fig. 3b), indicating the recruitment is dependent on ligand interaction with the $\beta_1AR$ orthosteric binding pocket. To further demonstrate recruitment of $G\alpha_i$ to carvedilol-stimulated $\beta_1AR$, we also performed co-immunoprecipitation experiments. Carvedilol stimulation increased the amount of $G\alpha_i$ bound to $\beta_1ARs$ in a dose-dependent manner, whereas it resulted in a decrease of $G\alpha_i$ that could be co-immunoprecipitated with $\beta_2ARs$ (Fig. 3c, Supplementary Fig. 3a). As a control for the effect of detergent on protein interaction during the co-immunoprecipitation, experiments were also performed with 1% n-Dodecyl $\beta$-D-maltoside (DDM) lysis buffer and showed similar results (Supplementary Fig. 3b). The amount of $G\alpha_i$ bound to $\beta_2ARs$ was increased by the balanced agonist isoproterenol (Fig. 3c), as we observed with the PLA experiments and again consistent with the previously identified process of $G\alpha_s/G\alpha_i$ switching[27].

We next determined whether carvedilol could induce $G\alpha_i$ protein activation using an antibody that specifically recognizes the active GTP-bound $G\alpha_i$. Carvedilol stimulation promoted the activation of $G\alpha_i$ in $\beta_1AR$ stable cells, but not in $\beta_2AR$ stable cells (Fig. 3d), which was blocked by PTX (Supplementary Fig. 3c).

To determine whether $G\alpha_i$ recruitment is specifically stimulated by carvedilol, we tested a number of $\beta AR$ agonists and antagonists with PLA (Fig. 4a) and co-immunoprecipitation (Fig. 4b, Supplementary Fig. 4). Remarkably, no other ligand tested induced $G\alpha_i$ recruitment to $\beta_1ARs$, suggesting that this process may be a unique property of the $\beta$-arrestin-biased ligand carvedilol.

Collectively, these data support a concept that carvedilol selectively promotes the recruitment and activation of $G\alpha_i$ to the $\beta_1AR$ subtype triggering $\beta$-arrestin-mediated signaling.

**Signaling dependence on both $G\alpha_i$ and $\beta$-arrestins.** Previous studies have shown that carvedilol stimulation of $\beta_1ARs$ promotes the internalization and activation of EGFRs, which in turn activates downstream signaling such as ERK phosphorylation[10]. To dissect the mechanism of carvedilol-induced $G\alpha_i$-dependent signaling, we tested the effect of PTX on $\beta_1AR$-mediated EGFR internalization. We transfected HEK293 cells stably expressing $\beta_1ARs$ with GFP-tagged EGFR, and monitored internalization by confocal microscopy. When stimulated with isoproterenol or carvedilol, GFP-EGFR redistributed from the plasma membrane into endosomes, similar to that observed after EGF treatment (Fig. 5a). Pretreatment with PTX significantly blocked the carvedilol-induced EGFR internalization, while without any effect on the isoproterenol response, indicating a requirement for $G\alpha_i$ for carvedilol-induced response (Fig. 5a).

Consistent with previous study showing that carvedilol-induced $\beta_1AR$-mediated EGFR transactivation is $\beta$-arrestin-dependent[10], siRNA knockdown of $\beta$-arrestin1 and $\beta$-arrestin2 abrogated both isoproterenol- and carvedilol-induced EGFR internalization (Fig. 5b, Supplementary Fig. 5a). While

transactivation triggered EGFR internalization induced by both ligands are $\beta$-arrestin dependent, the precise molecular mechanism appears to have distinct features. Whereas the carvedilol-induced response requires both $G\alpha_i$ and $\beta$-arrestin, the isoproterenol-induced response is PTX insensitive.

To more robustly determine the role of $G\alpha_i$ and $\beta$-arrestin in carvedilol-stimulated EGFR transactivation, we generated $\beta$-arrestin or $G\alpha_i$ deficient cells using CRISPR–Cas9 gene editing (Supplementary Fig. 5b). The wild type, $G\alpha_i$ knockout or $\beta$-arrestin1/2 knockout cells were transfected with CFP-tagged $\beta_1AR$. After ligand stimulation, the level of cell surface EGFRs was analyzed by flow cytometry (Fig. 5c). In wild-type cells, EGFRs were internalized following the treatment with EGF, isoproterenol or carvedilol. The depletion of $G\alpha_i$ blocked carvedilol-induced EGFR internalization, whereas absence of $G\alpha_i$ had no effect on EGF- or isoproterenol-induced responses. In contrast, $\beta$-arrestin1/2 knockout cells showed impaired EGFR internalization in response to both isoproterenol and carvedilol. Taken together, these results suggest that the carvedilol-induced EGFR internalization are dependent on both $G\alpha_i$ and $\beta$-arrestins.

Consistent with our observation for EGFR internalization, carvedilol-induced ERK phosphorylation required both $G\alpha_i$ and $\beta$-arrestins (Fig. 6a, b). Either $G\alpha_i$ inhibition by PTX or $\beta$-arrestin knockdown with siRNA diminished carvedilol-induced ERK phosphorylation (Fig. 6a). Moreover, in HEK293 cells transfected with FLAG-$\beta_1ARs$ but depleted of either $G\alpha_i$ or $\beta$-arrestin, carvedilol stimulated ERK activation was completely abrogated (Fig. 6b). Interestingly, removing either $\beta$-arrestin1 or $\beta$-arrestin2 prevented carvedilol-stimulated ERK phosphorylation, suggesting that both isoforms are required for carvedilol-stimulated signaling (Fig. 6b).

When $\beta ARs$ are stimulated by the balanced agonist isoproterenol, protein kinase A (PKA) activated by $G\alpha_s$-dependent cyclic AMP phosphorylates the receptor leading to a switch of $\beta_2AR$ G protein coupling from $G\alpha_s$ to $G\alpha_i$. The now $G\alpha_i$ coupled $\beta_2AR$ acts as a negative regulator of $G\alpha_s$ signaling and activates ERK signaling via dissociated $G\beta\gamma$ subunits from heterotrimeric $G\alpha_i$[27–29]. Here, we sought to determine if $G\beta\gamma$ subunits are required for carvedilol-stimulated $G\alpha_i$-dependent ERK phosphorylation. $G\beta\gamma$ inhibition was achieved by transfection of T8-$\beta ARKct$, a chimeric molecule consisting of two components: the C-terminus of the $\beta$ adrenergic receptor kinase ($\beta ARKct$) that competitively binds $G\beta\gamma$, therefore acting as an inhibitor of $G\beta\gamma$[30]; and the extracellular and transmembrane domain of CD8 receptor, which anchors the chimeric protein to the plasma membrane and potentiates its inhibitory effect[31]. The $G\beta\gamma$ blockade efficiency of T8-$\beta ARKct$ was confirmed by testing its effect on lysophosphatidic acid (LPA)-induced phosphorylation of cyclic AMP-responsive element-binding protein (CREB) (Fig. 7a). We show that the inhibition of $G\beta\gamma$ by T8-$\beta ARKct$ did not affect the carvedilol-induced ERK activation (Fig. 7b). This suggests that unlike isoproterenol stimulated $G\alpha_i$-signaling achieved by G

**Fig. 1** $G\alpha_i$ is required for the carvedilol-induced $\beta_1AR$-mediated ERK phosphorylation both in vitro and in vivo. **a** Effect of PTX on carvedilol-induced $\beta AR$-mediated ERK phosphorylation in HEK293 cells. HEK293 cells stably expressing FLAG-tagged $\beta_1ARs$ or $\beta_2ARs$ were pretreated with vehicle or 200 ng per ml PTX for 16 h, then stimulated with indicated concentration of carvedilol for 5 min. Carvedilol induced ERK phosphorylation in both $\beta_1AR$ or $\beta_2AR$ stable cells in dose-dependent manner. The response in $\beta_1AR$ stable cells was blocked by PTX, whereas that in $\beta_2AR$ stable cells was PTX insensitive. **b** Effect of $G\alpha_i$ knockout on $\beta AR$-mediated ERK phosphorylation in HEK293 cells. The $G\alpha_i$ expression in $\beta_1AR$ or $\beta_2AR$ stable cells was depleted with CRISPR–Cas9 gene editing. Compared with wild-type $\beta_1AR$ stable cells, the carvedilol-induced ERK phosphorylation in $G\alpha_i$ knockout $\beta_1AR$ stable cells was diminished. In comparison, the response in $\beta_2AR$ stable cells was not affected. **c** Effect of PTX on carvedilol-stimulated ERK phosphorylation in Langendorff perfused hearts from $\beta_2AR$ knockout mice or $\beta_1AR$ knockout mice. Mice were pretreated with vehicle or 25 μg per kg PTX through intraperitoneal injection. 48 h after injection, mice hearts were excised and perfused with vehicle or 10 μM carvedilol for 10 min. PTX diminished the carvedilol-induced ERK phosphorylation in hearts from $\beta_2AR$ knockout mice, but not $\beta_1AR$ knockout mice. Data represent the mean ± SEM for $n$ independent experiments (**a**, **b**) or $n$ animals (**c**) as marked on the figure. Statistical significance vs. control was assessed using two-way ANOVA (**a**, **b**) or one-way ANOVA (**c**) with Bonferroni correction. NS no stimulation; Carv carvedilol; Iso isoproterenol; p-ERK phosphorylated ERK; t-ERK total ERK; WT wild type; KO knockout

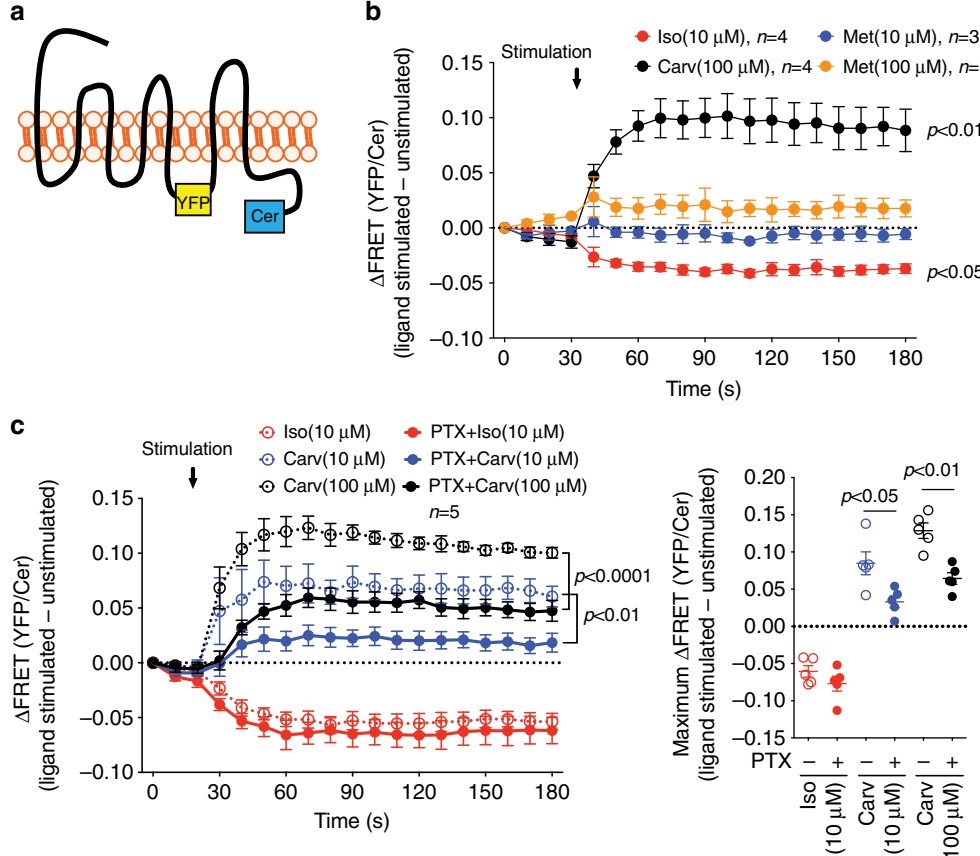

**Fig. 2** $G\alpha_i$ is required for carvedilol-induced $\beta_1AR$ conformation change. **a** In the FRET-based $\beta_1AR$ conformation sensor, YFP and Cerulean (Cer) are inserted in the third intracellular loop and the C-tail of $\beta_1AR$ respectively. **b** Ligand-induced changes of the FRET ratio in HEK293 cells stably expressing $\beta_1AR$ FRET sensor. The stable cells were stimulated with 100 µM carvedilol, 10 µM isoproterenol, 10 µM or 100 µM $\beta_1AR$ antagonist metoprolol, while FRET was monitored in real-time as the emission ratio of YFP to Cer. Carvedilol stimulation increased the FRET ratio, while isoproterenol decreased it, demonstrating the distinct $\beta_1AR$ conformations induced by these two ligands. Metoprolol had no effect on the FRET ratio. **c** Effect of PTX on ligand-induced FRET ratio change. Cells were pretreated with vehicle or 200 ng per ml PTX for 16 h before ligand stimulation. PTX blocked carvedilol-induced change, while having no effect on the isoproterenol response, suggesting that $G\alpha_i$ is required to stabilize the carvedilol-induced $\beta_1AR$ conformation. Data represent the mean ± SEM for $n$ independent experiments as marked on the figure. Statistical significance vs. unstimulated cells was assessed using one-way ANOVA with Bonferroni correction (**b**); statistical significance between PTX-pretreated and non-pretreated cells was assessed using two-way ANOVA with Bonferroni correction (**c**, left panel), or two-tailed paired Student's $t$-test (**c**, right panel)

protein switching, carvedilol-induced $G\alpha_i$-dependent $\beta_1AR$ signaling does not require $G\beta\gamma$.

Collectively, these data demonstrate that both $G\alpha_i$ and $\beta$-arrestins are required for carvedilol-induced $\beta_1AR$ signaling. Notably, either PTX pretreatment or $\beta$-arrestin knockdown was able to significantly block the carvedilol-induced $\beta_1AR$-mediated EGFR internalization and ERK phosphorylation, and these responses were completely abrogated when either $G\alpha_i$ or $\beta$-arrestin was depleted by gene editing. Taken together these data suggest that $G\alpha_i$ and $\beta$-arrestins are likely involved in the same signaling cascade, rather than acting in parallel pathways downstream of $\beta_1AR$.

**Phosphorylation of $\beta_1AR$ is not required for $G\alpha_i$ recruitment**. GRK-mediated receptor phosphorylation plays a critical role in $\beta$-arrestin-dependent signaling of $\beta ARs$[32]. When $\beta_1ARs$ are stimulated by balanced agonists, such as isoproterenol or dobutamine, GRK-mediated $\beta_1AR$ phosphorylation of the carboxyl-terminal tail occurs and is required for agonist mediated $\beta$-arrestin recruitment[9, 33]. For the $\beta_2AR$, a similar process has been show to occur whereby stimulation with the $\beta$-arrestin-biased agonist carvedilol promotes $\beta_2AR$ phosphorylation at specific

GRK sensitive amino acid residues 355 and 356 of the c-terminal tail[21, 32]. Here, we sought to determine whether GRK-mediated phosphorylation of the $\beta_1AR$ is a necessary step in the carvedilol-induced $G\alpha_i$ recruitment to the receptor. To address this question, we used a mutant $\beta_1AR$ that lacks the putative GRK phosphorylation sites within the receptor carboxyl-terminal tail (GRK-$\beta_1AR$) and therefore unable to be phosphorylated by GRKs[9, 33]. We show that carvedilol stimulation increased $G\alpha_i$ recruitment to a similar extent between wild type and GRK-$\beta_1ARs$, as assessed by co-immunoprecipitation (Fig. 8a) and suggests that GRK-mediated $\beta_1AR$ phosphorylation is not required for carvedilol-induced $G\alpha_i$ recruitment to the $\beta_1AR$.

$\beta ARs$ can switch coupling from $G\alpha_s$ to $G\alpha_i$ when stimulated with a balanced agonist[27, 29]. In the $G\alpha_s$-$G\alpha_i$ switching model, agonist stimulated $\beta_2AR$–$G\alpha_i$ coupling is dependent on PKA-mediated receptor phosphorylation[27, 29]. To determine whether a similar mechanism is involved in the carvedilol-induced $G\alpha_i$ recruitment to $\beta_1ARs$, we used a $\beta_1AR$ mutant lacking the putative PKA phosphorylation sites (PKA-$\beta_1AR$) or the PKA inhibitor H89. In our experiments, carvedilol stimulation promotes the $G\alpha_i$ recruitment to PKA-$\beta_1ARs$, similar as to wild-type receptors (Fig. 8a), whereas PKA inhibition with H89 did not have a significant effect (Fig. 8b).

Taken together, these data suggest that neither GRK- nor PKA-mediated receptor phosphorylation is required for carvedilol-induced $G\alpha_i$ recruitment to $\beta_1ARs$.

**$\beta_1AR$ C-tail is required but not sufficient for $G\alpha_i$ coupling.** Since the C-terminus of the βARs play vital roles in recruiting signal effectors and regulating downstream signaling[34], we postulated that specific amino acid residues within the $\beta_1AR$ C-tail

are critical for receptor subtype specificity of $G\alpha_i$ recruitment. To test this hypothesis, we transfected HEK293 cells with βAR chimera mutants in which the C-tail of $\beta_1ARs$ was exchanged with that of $\beta_2ARs$[14], and assessed $G\alpha_i$ recruitment to chimera βARs with co-immunoprecipitation. Carvedilol stimulation promoted the recruitment of $G\alpha_i$ to the wild-type $\beta_1ARs$, but was abrogated when the $\beta_1AR$ contained the C-tail from the $\beta_2AR$ ($\beta_{1/2}AR$) (Fig. 8c). In contrast, the effect of carvedilol on $G\alpha_i$ recruitment to the $\beta_2AR$ with the $\beta_1AR$ C-tail ($\beta_{2/1}AR$) was similar to that of

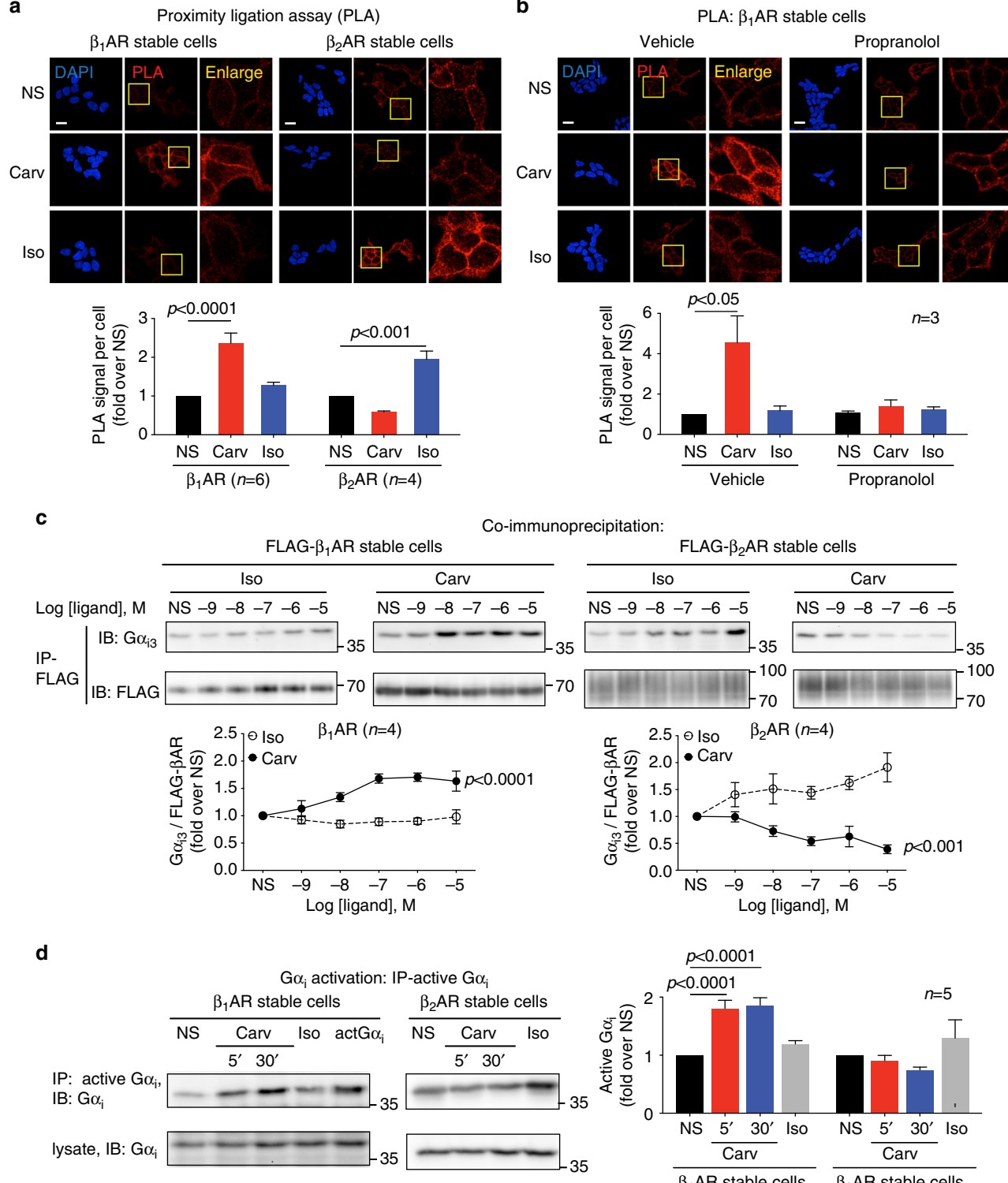

wild-type β$_2$ARs. These data suggest that the C-tail of the β$_1$AR is required for Gα$_i$ recruitment, but alone is insufficient for this process to occur and is consistent with the crystal structure of the β$_2$AR and G protein complex showing multiple receptor-G protein contact points[35].

## Discussion

In this study, we provide new insight into the molecular mechanism of biased agonism at the β$_1$AR. Carvedilol, a ligand classically known as a βAR antagonist, activates β-arrestin signaling by switching the uniquely Gα$_s$-coupled β$_1$AR to a Gα$_i$-coupled receptor. We show that carvedilol is unique among a number of agonists and antagonists tested to promote the recruitment and activation of Gα$_i$ to β$_1$ARs. The recruited Gα$_i$ in turn stabilizes a carvedilol-bound β$_1$AR conformation that is required for β-arrestin-biased β$_1$AR signaling as measured by EGFR internalization and ERK phosphorylation. These results indicate that the previously defined G protein bias vs. β-arrestin-bias may be attributed to ligand-induced selective coupling of receptors to specific G protein subtypes, i.e., G protein subtype bias. In our conceptual model for β$_1$AR biased signaling, we speculate that binding of carvedilol to the β$_1$AR stabilizes a unique receptor conformation that recruits and activates Gα$_i$ to promote β-arrestin-mediated signaling (Fig. 9). While carvedilol is also known to stimulate β$_2$AR signaling, Gα$_i$ recruitment was not required for β$_2$AR-mediated β-arrestin-biased signaling and suggests that different mechanisms for bias may be operative between βAR subtypes.

In the classical view of GPCR signaling, agonist stabilization of specific active conformational states promotes coupling of heterotrimeric G proteins and stimulation of downstream signaling[36]. Receptor signaling is then terminated by a process involving receptor phosphorylation, β-arrestin recruitment and receptor internalization. However, recent studies suggest that the classical "on–off" (active and inactive) model is oversimplified[20], as GPCRs transmit signaling through multiple transducers to regulate diverse arrays of pathways. First, some GPCRs can couple to multiple G proteins. For example, the isoproterenol-activated β$_2$AR switches coupling from Gα$_s$ to Gα$_i$[27]. In this study, we show that carvedilol switches the classical Gα$_s$-coupled receptor β$_1$AR to a Gα$_i$-coupled receptor. However, in contrast to the Gα$_s$-Gα$_i$ switching of the β$_2$AR, the carvedilol-induced β$_1$AR–Gα$_i$ coupling does not involve Gα$_s$ activation and PKA-mediated receptor phosphorylation. The carvedilol-induced β$_1$AR–Gα$_i$ signaling is also different from the actions of classical Gα$_i$-coupled receptors such as the muscarinic M$_2$ receptor and the α$_2$ adrenergic receptor[37], as its activation of ERK is not mediated through Gβγ subunits. Second, in addition to their role as signal terminators for G protein signaling, β-arrestins can act as signal transducers in their own right. Current conceptual

models support the idea that ligands may differentially stabilize distinct receptor conformations that recruit divergent portfolio of signaling transducers and effector proteins to active a select suite of cellular signaling pathways, a concept termed functional selectivity or biased agonism[15].

The β-arrestin-biased ligand carvedilol has three unique features at the β$_1$AR: (1) it has inverse efficacy for Gα$_s$-dependent adenylyl cyclase activity; (2) it promotes the recruitment of Gα$_i$, not Gα$_s$, to the β$_1$AR; (3) it activates the classical β-arrestin signaling using a Gα$_i$ paradigm. These unique signaling properties of carvedilol may be attributed to its ability to stabilize a distinct receptor active conformation[15]. For the β$_2$AR, carvedilol uniquely induces significant conformational rearrangement around residue Lys263 and Cys265 in the third intracellular loop of the receptor, which may expose the loop toward intracellular surface and facilitate the receptor interaction with β-arrestins[15]. Though a previous study suggests the crystal structure of carvedilol-bound β$_1$AR is similar to that of the cyanopindolol-bound inactive state structure[38], additional conformations stabilized by carvedilol may require the binding of transducers such as Gα$_i$ or β-arrestin. This requirement of transducer binding for receptor conformational stability is supported by the structural study of the β$_2$AR showing that the interaction of a G protein, or a G protein-like-protein nanobody, is required to stabilize the agonist-induced receptor active conformation[39]. In our study, using a FRET-based β$_1$AR conformation sensor, we show that carvedilol induces a change of FRET ratio, representing a receptor conformational change. Notably, the β$_1$AR conformation induced by carvedilol is distinct from the one induced by the balanced agonist isoproterenol, as carvedilol increased the FRET ratio while isoproterenol decreased it. This further supports a concept that receptors can adopt distinct conformations when stimulated by different ligands. As our results show that carvedilol promotes the recruitment of Gα$_i$ to β$_1$ARs, while a wide range of other βAR ligands tested do not, it is possible that carvedilol induces a β$_1$AR conformational change that exposes allosteric binding sites on the receptor to allow for receptor–Gα$_i$ interaction. In turn, the bound Gα$_i$ stabilizes the carvedilol-induced active receptor conformation and is consistent with our data where pretreatment with the Gα$_i$ inhibitor PTX impairs the carvedilol activated β$_1$AR conformation. Together these data support the concept that carvedilol-induced Gα$_i$ is a positive allosteric modulator of the β-arrestin-biased β$_1$AR active conformation.

While we have not determined the precise mechanism of how Gα$_i$ binding to the carvedilol-occupied β$_1$AR triggers β-arrestin signaling, we postulate that it may involve subsequent receptor phosphorylation in a process known as the "barcode" hypothesis[19]. Upon ligands stimulation, GPCRs can be phosphorylated by distinct GRK subtypes at specific sites. Previous study identified β$_2$AR sites that are specifically phosphorylated by GRK2 and GRK6[32]. While the balanced agonist isoproterenol stimulates

**Fig. 3** Carvedilol promotes Gα$_i$ recruitment and activation in β$_1$AR stable cells, but not in β$_2$AR stable cells. HEK293 cells stably expressing FLAG-tagged β$_1$ARs or β$_2$ARs were stimulated with 10 μM carvedilol or 10 μM isoproterenol for 5 min. **a** In proximity ligation assay (PLA), cells were immuno-stained with Gα$_i$ antibody raised in mouse and β$_1$AR (or β$_2$AR) antibody raised in rabbit. The red PLA signal represents the protein interactions of Gα$_i$ and β$_1$AR (or β$_2$AR). The area in yellow squares are enlarged for better view. Carvedilol promoted Gα$_i$ recruitment to β$_1$ARs, but not to β$_2$ARs. Scale bar = 20 μm. **b** β$_1$AR stable cells were pretreated with vehicle or 10 μM propranolol for 30 min. The βAR antagonist propranolol blocked the carvedilol response, suggesting that β$_1$AR–Gα$_i$ coupling is induced by the binding of carvedilol to the β$_1$AR orthosteric binding pocket. Scale bar = 20 μm. **c** The effect of carvedilol and isoproterenol on Gα$_i$ recruitment was confirmed with co-immunoprecipitation assays. FLAG-tagged β$_1$ARs or β$_2$ARs were immunoprecipitated with anti-FLAG M2 beads, and Gα$_{i3}$ was detected with its specific antibody by western blot. **d** Carvedilol specifically activated Gα$_i$ in β$_1$AR stable cells. Cells were transfected with Gα$_{i2}$. 48 h after transfection, cells were treated with 10 μM carvedilol for 5 min or 30 min, or 10 μM isoproterenol for 5 min. Activated Gα$_i$ was immunoprecipitated with an antibody specifically recognizing the active form of Gα$_i$, and immunoblotted with an Gα$_i$ antibody. In the middle lane marked as actGα$_i$, cells were transfected with constitutively active Gα$_{i2}$, serving as positive control. Carvedilol stimulation activated Gα$_i$ in β$_1$AR stable cells, but not in β$_2$AR stable cells. Data represent the mean ± SEM for n independent experiments as marked on the figure. Statistical significance vs. unstimulated cells (NS) was assessed using one-way ANOVA with Bonferroni correction

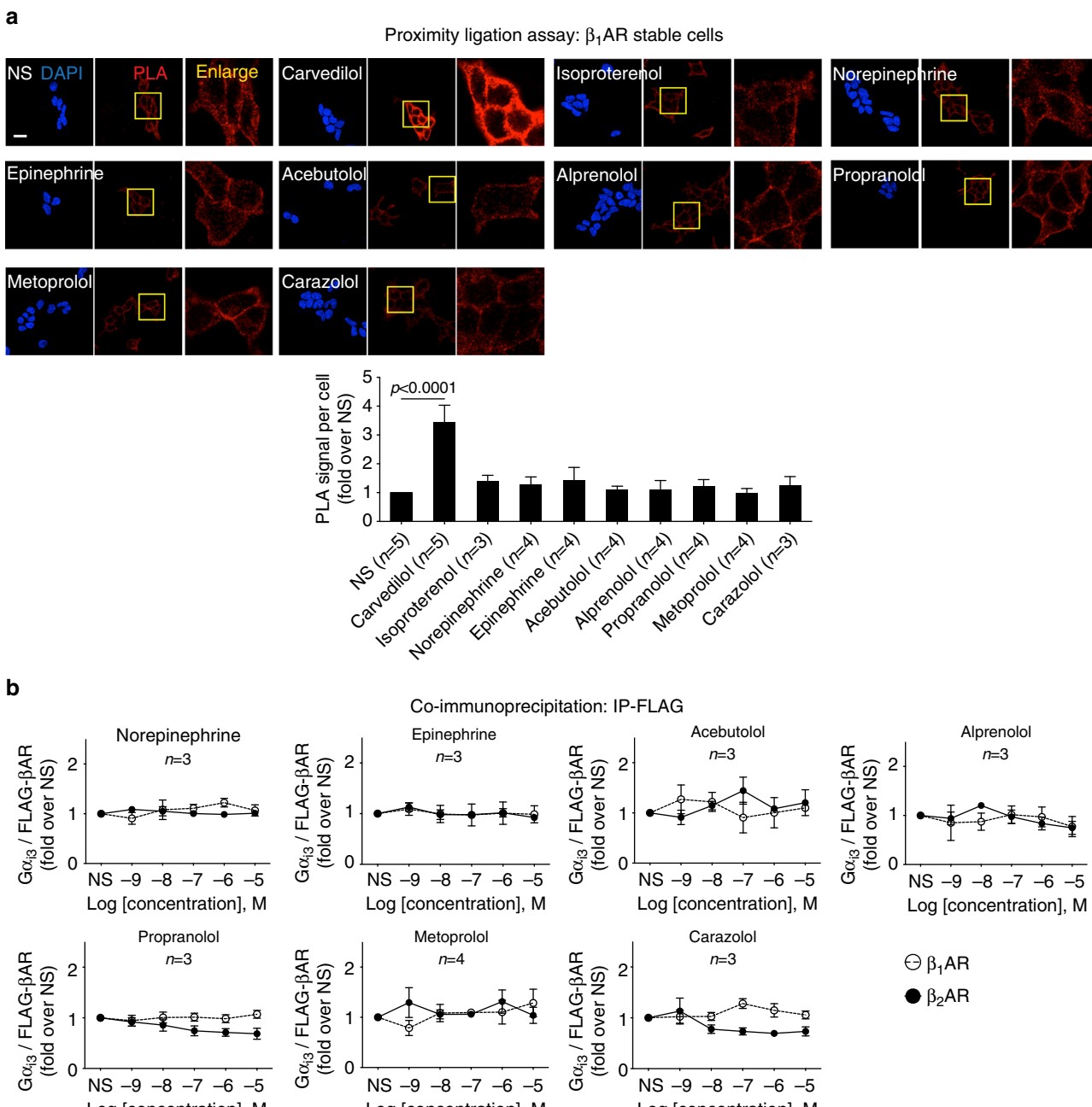

**Fig. 4** A number of βAR agonists or antagonists tested do not have significant effect on Gα$_i$ recruitment. **a** β$_1$AR stable cells were stimulated with vehicle or 10 μM indicated ligands for 5 min. Interaction of β$_1$AR and Gα$_i$ were detected by PLA. Scale bar = 20 μm. **b** β$_1$AR or β$_2$AR stable cells were stimulated with ligands at indicated concentration for 5 min. Gα$_i$ recruitment was detected by co-immunoprecipitation. Both assays suggested that none of the ligands tested had similar effect of carvedilol on Gα$_i$ recruitment. Data represent the mean ± SEM for $n$ independent experiments as marked on the figure. Statistical significance vs. unstimulated cells was assessed using one-way ANOVA with Bonferroni correction

β$_2$AR phosphorylation at both GRK2- and GRK6-specific sites, carvedilol only stimulates receptor phosphorylation at the GRK6-specific sites. This "barcode" phosphorylation pattern of receptors plays essential roles in regulating the recruitment and functionality of signaling transducers[19]. For instance, β$_2$AR phosphorylation mediated by GRK2 and GRK6 induces distinct β-arrestin conformations, and differentially regulates receptor internalization and ERK activation[32]. Similarly for the β$_1$AR, GRK2-mediated and GRK5/6-mediated receptor phosphorylation leads to distinct cellular responses[9, 40], suggesting that a phosphorylation barcode for the β$_1$AR may also direct β-arrestin signaling. To dissect the mechanism of how Gα$_i$ regulates β$_1$AR signaling,

future studies will need to compare the isoproterenol- or carvedilol-induced barcode phosphorylation patterns of the β$_1$AR, as well as the effect of Gα$_i$ inhibitor PTX on it.

While our data show that both Gα$_i$ and β-arrestins are required for carvedilol-induced biased signaling of the β$_1$AR, whether β-arrestin is recruited to the carvedilol occupied β$_1$AR remains to be determined. Using a number of methodologies, such as co-immunoprecipitation, confocal- or bioluminescence resonance energy transfer-based assays, we were unable to detect carvedilol-induced β-arrestin recruitment to the β$_1$AR. This may be due to a number of reasons: (1) ligand-induced β-arrestin recruitment and activation is rapid, within 2 s after stimulation, and reversible[41];

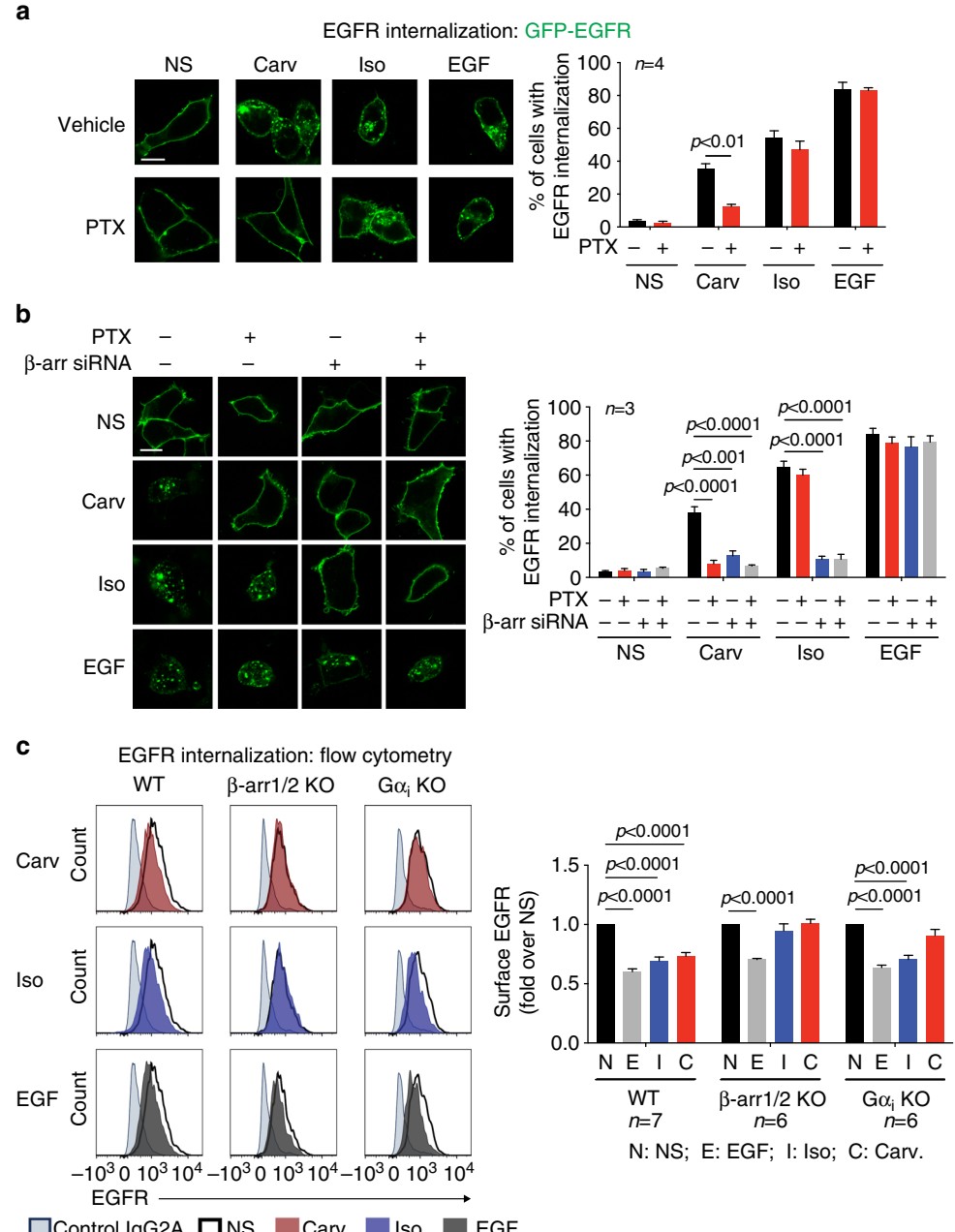

**Fig. 5** Both $G\alpha_i$ and $\beta$-arrestins are required for carvedilol-induced $\beta_1AR$-mediated EGFR internalization. **a** The effect of PTX on ligand-stimulated EGFR internalization. $\beta_1AR$ stable cells with transient transfection of GFP-EGFR were pretreated with vehicle or 200 ng per ml PTX for 16 h. Then the cells were stimulated with 10 μM carvedilol, 10 μM isoproterenol or 10 ng per ml EGF for 5 min. Both carvedilol and isoproterenol promoted EGFR internalization, but only the carvedilol-induced response was PTX sensitive. Scale bar = 10 μm. **b** Either PTX pretreatment or $\beta$-arrestins knockdown blocked carvedilol-induced EGFR internalization. $\beta_1AR$ stable cells were transfected with GFP-EGFR together with scrambled control siRNA or $\beta$-arrestin1/2 siRNA. 48 h after transfection, cells were pretreated with vehicle or 200 ng per ml PTX for 16 h before stimulation. Scale bar = 10 μm. **c** Carvedilol-induced EGFR internalization was abrogated in $\beta$-arrestins or $G\alpha_i$ knockout cells. Wild type, $\beta$-arrestin1/2 knockout or $G\alpha_i$ knockout cells were transfected with CFP-tagged $\beta_1ARs$. Cells were stimulated with 10 μM carvedilol, 10 μM isoproterenol or 10 ng per ml EGF for 5 min. The EGFR level on cell surface was assessed by flow cytometry. Both carvedilol- and isoproterenol-induced EGFR internalization were impaired in the $\beta$-arrestin knockout cells, whereas only the carvedilol-induced response was blocked in the $G\alpha_i$ knockout cells. Data represent the mean ± SEM for $n$ independent experiments as marked on the figure. Statistical significance was assessed using two-tailed paired Student's $t$-test (**a**) or one-way ANOVA with Bonferroni correction (**b**, **c**)

(2) the affinity of the $\beta_1AR$–$\beta$-arrestin interaction is low. Both the $\beta_1AR$ and the $\beta_2AR$ are known as class A receptors, since they are characterized by transient and weak interaction with $\beta$-arrestins along with a rapid recycling to the plasma membrane after internalization. To demonstrate carvedilol triggered $\beta$-arrestin recruitment to the $\beta_2AR$, previous studies used a chimeric receptor consisting of the $\beta_2AR$ fused to vasopressin $V_2$ receptor cytoplasmic tail ($\beta_2AR$-$V_2R$) to increase the affinity of $\beta$-arrestin to the ligand occupied receptor[21]. However, as we have shown (Fig. 8c), the C-tail of the $\beta_1AR$ is required for $G\alpha_i$ recruitment. Therefore substituting the $\beta_1AR$ C-tail with the $V_2R$ tail would not provide a chimeric receptor suitable to study the role of $G\alpha_i$ in carvedilol stimulated $\beta$-arrestin recruitment. Importantly, we cannot exclude that the carvedilol-stimulated $\beta_1AR$ signaling is

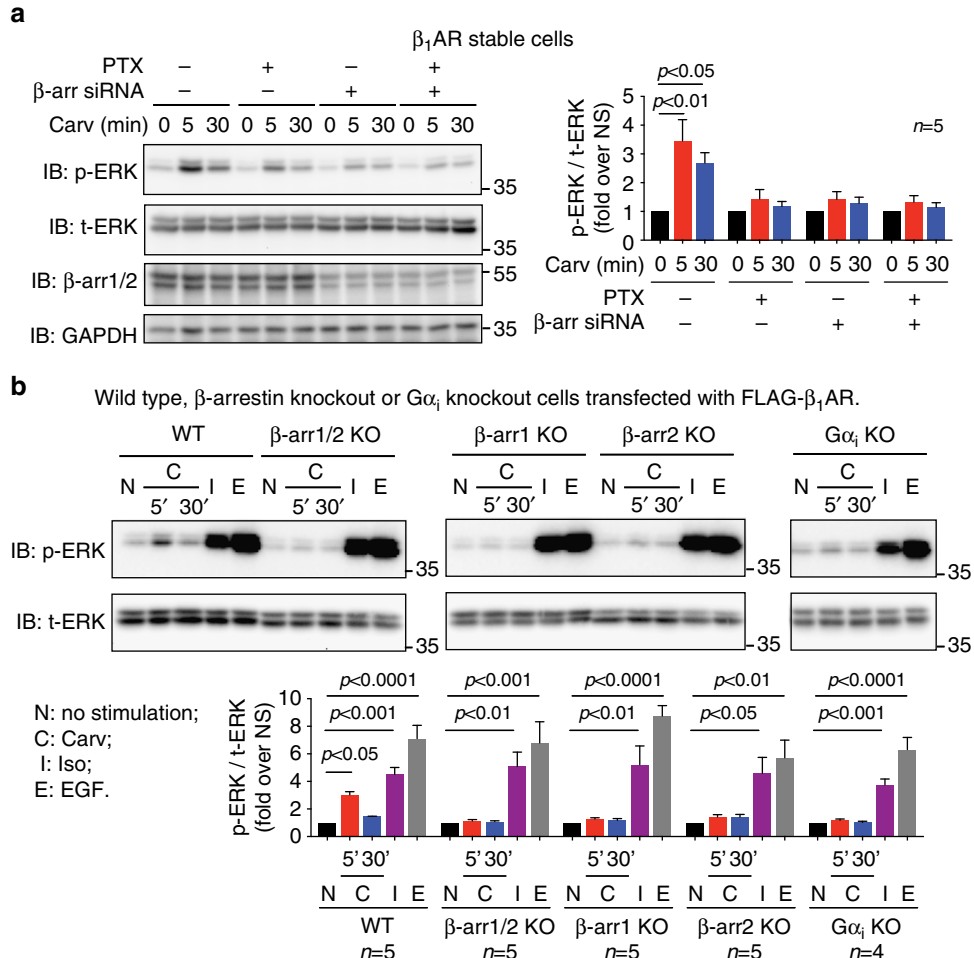

**Fig. 6** Carvedilol-induced β₁AR-mediated ERK phosphorylation is dependent on both Gα$_i$ and β-arrestins. **a** The effect of PTX and β-arrestin knockdown on carvedilol-stimulated ERK phosphorylation. β₁AR stable cells with transfection of control siRNA or β-arrestin1/2 siRNA were pretreated with vehicle or PTX, then simulated with 10 µM carvedilol for 5 min or 30 min. Carvedilol-stimulated ERK phosphorylation was diminished by either PTX pretreatment or β-arrestins siRNA, suggesting the requirement of Gα$_i$ and β-arrestins for this signaling. **b** The β₁AR-mediated ERK phosphorylation in β-arrestin or Gα$_i$ knockout cells. Wild type, β-arrestin knockout or Gα$_i$ knockout HEK293 cells were transfected with FLAG-tagged β₁ARs. Cells were stimulated with 10 µM carvedilol for 5 or 30 min, 10 µM isoproterenol or 10 ng per ml EGF for 5 min. The depletion of either β-arrestins or Gα$_i$ impaired carvedilol-induced ERK phosphorylation. Data represent the mean ± SEM for $n$ independent experiments as marked on the figure. Statistical significance vs. unstimulated cells was assessed using one-way ANOVA with Bonferroni correction

mediated by β-arrestin by an indirect mechanism that does not require direct binding of β-arrestin to the β₁AR. A recent study identified unique features for the β₁AR with respect to β-arrestin interaction and activation[42], where a brief interaction with the activated β₁AR is sufficient to target β-arrestin2 to clathrin-coated structures and trigger ERK signaling even in the absence of receptor association[42]. This β-arrestin "activation at a distance" mechanism suggests that a β₁AR–β-arrestin complex may not be essential for the activation of β-arrestin-dependent signaling and could explain our findings for a role of β-arrestin in carvedilol-induced signaling without a direct β₁AR–β-arrestin interaction. Lastly, it is also possible that instead of directly engagement with the β₁AR, β-arrestins could associate with other components of the signaling cascade such as the transactivated EGFR. This has recently been shown for the vasopressin V₂ receptor signaling, where β-arrestins are recruited to, and act downstream of, the transactivated insulin-like growth factor receptor[43].

Carvedilol is a βAR antagonist (β-blocker), a family of drugs that are widely used in the therapeutic treatment of cardiovascular diseases such as hypertension and heart failure, as β₁ARs and β₂ARs are predominant GPCR subtypes expressed in

mammalian heart and play vital roles in the regulation of cardiac function[4]. In heart failure, treatment with β-blockers improves left ventricle function, reverses the pathological cardiac remodeling, and reduces mortality and morbidity[44, 45]. However, β-blockers have different clinical efficacies. Some evidence suggests that carvedilol has a superior effect on cardiovascular survival to other β-blockers[46]. The molecular basis for this remains to be elucidated, but has been attributed to the additional properties of carvedilol other than β-blockers, such as the antioxidant, antiproliferative effects and α₁ adrenergic receptor blockade[47]. Interestingly, carvedilol appears to be unique among βAR blockers in that it can activate β-arrestin-dependent signaling that confers cardioprotection[10, 21]. Given the possible cardioprotective role of Gα$_i$ during cardiac stress[48] and the ability of carvedilol to promote β₁AR–Gα$_i$ coupling, it is possible that this unique property of carvedilol is also important for its therapeutic efficacy.

In conclusion, we identify a new signaling mechanism of GPCR biased agonism. To date, the β₁AR was considered to be predominantly coupled to Gα$_s$, and β-arrestin-dependent β₁AR signaling to be independent of G proteins. However, our data supports a concept where carvedilol has three unique properties

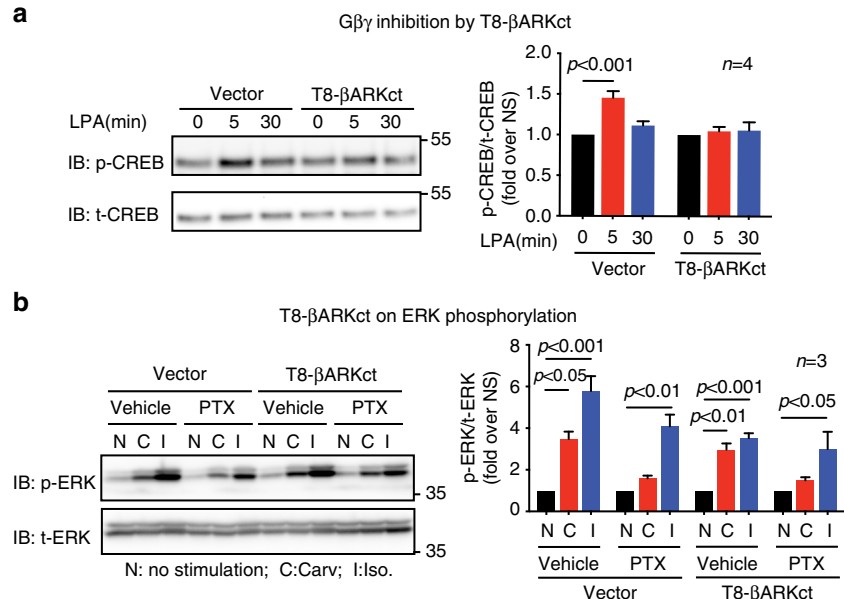

**Fig. 7** Gβγ subunits are not required for carvedilol-induced β$_1$AR-mediated ERK phosphorylation. **a** Validation of the Gβγ inhibition by T8-βARKct. HEK293 cells with or without transient transfection of T8-βARKct were stimulated with 10 μM LPA for 5 min. T8-βARKct diminished the LPA-induced CREB phosphorylation, a known Gβγ-dependent process, confirming the inhibition of Gβγ subunits by T8-βARKct. **b** The Gβγ subunits are not required for carvedilol-induced β$_1$AR-mediated ERK phosphorylation. β$_1$AR stable cells with or without T8-βARKct transfection was pretreated with vehicle or 200 ng per ml PTX for 16 h. The cells were then stimulated with 10 μM carvedilol or 10 μM isoproterenol for 5 min. T8-βARKct did not have significant effect on ERK phosphorylation, suggesting that Gβγ subunits were not required. Data represent the mean ± SEM for $n$ independent experiments as marked on the figure. Statistical significance vs. unstimulated cells was assessed using one-way ANOVA with Bonferroni correction

at the β$_1$AR: (1) it is inert with respect to Gα$_s$; (2) it recruits Gα$_i$ and converts the β$_1$AR from a Gα$_s$-coupled receptor to one that couples to Gα$_i$; and (3) it activates classical β-arrestin-dependent signaling in a Gα$_i$ paradigm. These data suggest a greater complexity for receptor signaling bias than previously appreciated in that coupling of distinct G protein subtypes to the activated receptor are needed for β-arrestin-biased agonism. These data also have important implications when considering the development of new therapeutic ligands designed to selectively target β-arrestin-biased signaling pathways.

## Methods

**Cell culture**. HEK293 cells (American Type Culture Collection) stably expressing FLAG-tagged β$_1$AR or β$_2$AR are maintained and transfected as previously described[33, 49]. Cells were periodically treated with BMCyclin (Roche) to avoid mycoplasma contamination. Cells were incubated overnight in serum-free medium supplemented with 0.1% BSA, 10 mM HEPES and 1% penicillin–streptomycin and pretreated with pertussis toxin (200 ng per ml, overnight), H89 (10 μM, 30 min) or propranolol (10 μM, 30 min) before ligand stimulation. HEK293 cells stably expressing β$_1$AR-FRET sensor were used for the FRET experiments.

**Generation of β-arrestin or Gα$_i$ knockout cell line**. Plasmids carrying *S. pyogenes* Cas9 (SpCas9) next to a cloning site for guide RNA (gRNA) with EGFP (pSpCas9 (BB)-2A-GFP, Addgene 48138) or puromycin resistant gene (pSpCas9(BB)-2A-Puro, Addgene 48139) were obtained from Addgene (deposited by the laboratory of Dr. F. Zhang[50]). Designing of the guide RNAs for Gα$_i$ or β-arrestins and cloning the guide RNAs into the Cas9 plasmids were performed as previously described[50].
For β-arrestin knockout cells, β-arrestin1 was targeted using guide sequence oligos (top: CACCGCATCGACCTCGTGGACCCTG; bottom: AACCAGGGTCCACGAGGTCGATGC). β-arrestin2 was targeted using guide sequence oligos (top: CACCGCGTAGATCACCTGGACAAAG; bottom: AAACCTTTGTCCAGGTGATCTACGC). The guide sequence oligos were cloned into pSpCas9(BB)-2A-Puro. After confirming the cloning by sequencing, plasmids were transfected into HEK293 cells using Fugene 6 transfection reagent (Promega). 72 h after transfection, cells were harvested to check INDEL (insertion deletion) in the genome by surveyor's assay. Puromycin (2.5 μg per ml) was added into the medium of surveyor positive cells to select cells with the plasmid containing puromycin resistant gene along with guide RNA and Cas9. The knockout of β-arrestins were confirmed by western blot.

For Gα$_i$ knockout cells, Gα$_{i1}$ was targeted using guide sequence oligos (top: CACCGCGCCGTCCTCACGGAGGTTG; bottom: AAACCAACCTCCGTGAGGACGGCGC), Gα$_{i2}$ was targeted using guide sequence oligos (top: CACCGAGACAACCGCCCGGTACTGC, bottom: AAACGCAGTACCGGGCGGTTGTCTC), and Gα$_{i3}$ was targeted using guide sequence oligos (top: CACCGGACGGCTAAAGATTGACTT; bottom: AAACAAGTCAATCTTTAGCCGTCCC). The guide sequence oligos were cloned into pSpCas9 (BB)-2A-GFP. Plasmids targeting the three Gα$_i$ subtypes were co-transfected into HEK293 cells. GFP positive cells were selected by fluorescence-activated cell sorting, diluted for growth and single cell colonies were obtained. The Gα$_i$ knockout were confirmed by western blot.

**Immunoblotting and immunoprecipitation**. Following stimulation, cells were scraped in 1% NP-40 lysis buffer (20 mM Tris, pH 7.4, 137 mM NaCl, 20% glycerol, 1% Nonidet P-40, 2 mM sodium orthovanadate, 1 mM PMSF, 10 mM sodium fluoride, 10 μg per ml aprotinin, 5 μg per ml leupeptin and phosphatase inhibitors) or 1% DDM lysis buffer (20 mM HEPES, 150 mM NaCl, 1% n-Dodecyl β-ᴅ-maltoside, protease inhibitors and phosphatase inhibitors). For immunoprecipitation of FLAG-tagged β$_1$AR or β$_2$AR, 1–2 mg of protein was incubated overnight with 30 μl of anti-FLAG M2 magnetic beads (Sigma). For immunoprecipitation of active Gα$_i$, protein was incubated for 2 h with anti-active Gα$_i$ antibody (New East Biosciences) and Protein A/G beads (EMD Millipore). Immunoprecipitates or cell lysate samples were separated by SDS-PAGE, transferred to PVDF membrane (Bio-Rad) and subjected to immunoblotting with various primary antibodies. Immunoblots were detected using enhanced chemiluminescence (Thermo Fisher Scientific) and analyzed with ImageJ software. Uncropped blots are shown in Supplementary Fig. 6.

**Antibodies**. Please refer the information of antibodies to Supplementary Table 1.

**ERK phosphorylation in mice heart**. Eight to 12-week-old gender-matched β$_1$AR knockout (β$_1$AR KO) mice and β$_2$AR KO mice[51] were used for this study. Three to six animals were used for each experimental group based on previous experiments. Randomization and blinding were not performed. Mice were pretreated with vehicle or 25 μg per kg pertussis toxin (PTX) via intraperitoneal injection. After 48 h, mice were anesthetized with ketamine (100 mg per kg) and xylazine (2.5 mg per kg) for 10 min. Heart was then excised and, with aorta cannulated to needle, perfused with perfusion buffer (118 mM NaCl, 4.7 mM KCl, 1.2 mM MgSO$_4$, 1.2 mM KH$_2$PO$_4$, 2.5 mM CaCl$_2$, 25 mM NaHCO$_3$, 0.5 mM Na-EDTA, 5.5 mM glucose) with O$_2$ bubbling through Langendorff apparatus (Hugo Sachs Harvard Apparatus) set at 37 °C. After 10 min perfusion, buffer was changed to perfusion buffer with vehicle or 10 μM carvedilol, and perfused for another 10 min. Heart was

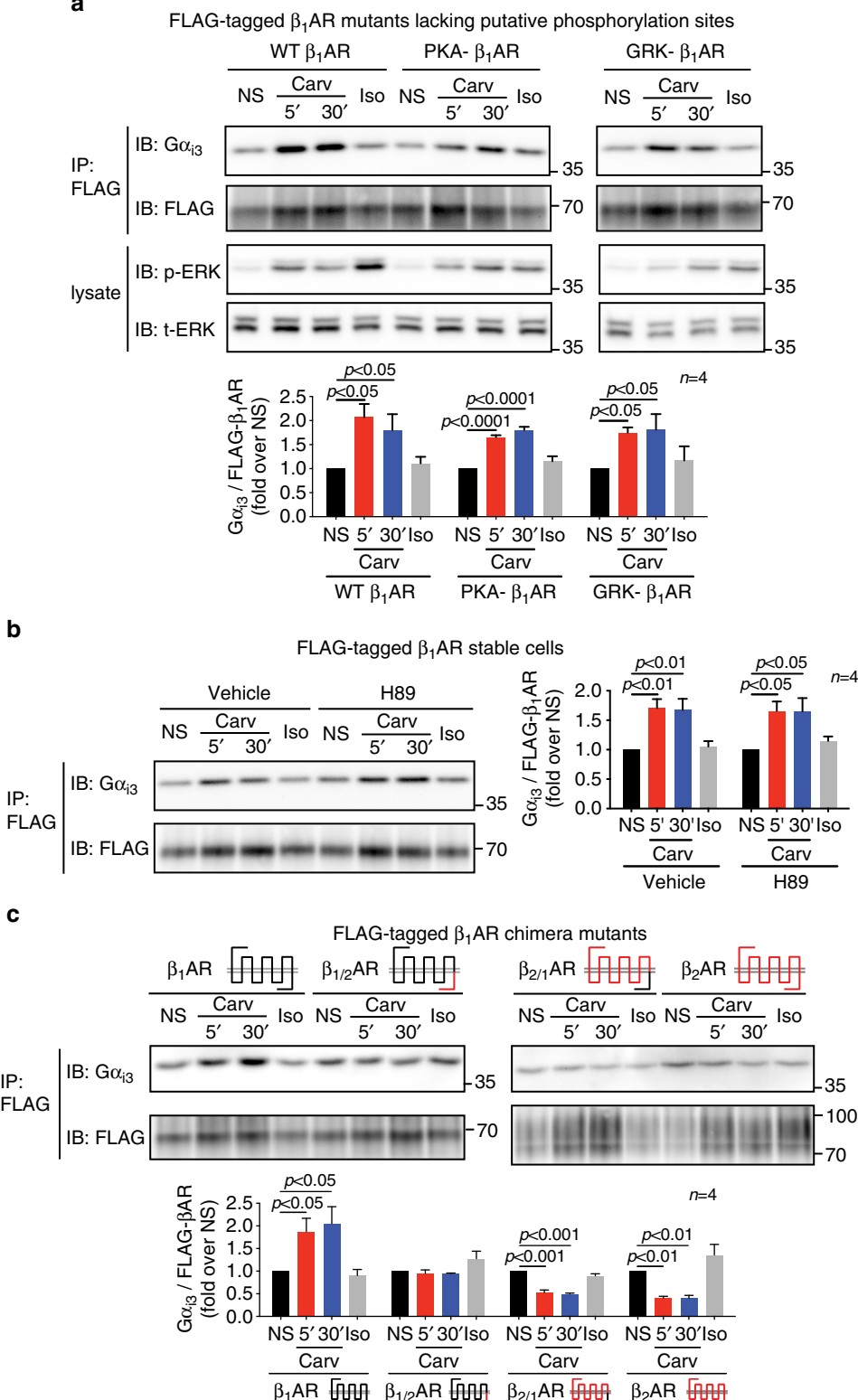

**Fig. 8** Neither PKA-mediated nor GRK-mediated β₁AR phosphorylation is required for carvedilol-induced Gα$_i$ recruitment. C-tail of β₁AR is required but not sufficient for Gα$_i$ recruitment. **a** HEK293 cells were transfected with FLAG-tagged wild-type, PKA- or GRK- β₁ARs. Carvedilol promoted Gα$_i$ recruitment to mutant β₁ARs lacking the putative PKA- or GRK-mediated phosphorylation sites, to a similar extent as to the wild-type β₁ARs. **b** HEK293 cells stably expressing FLAG-tagged β₁ARs were pretreated with vehicle or 10 μM H89 for 30 min. The PKA inhibitor H89 did not have a significant effect on carvedilol-induced β₁AR–Gα$_i$ coupling. **c** HEK293 cells were transfected with FLAG-tagged β₁AR, β₂AR or chimeric βAR constructs in which the receptor C-tails were exchanged between the two receptor subtypes. Carvedilol did not promote Gα$_i$ recruitment to the β₁AR with C-tail from β₂AR. On the other hand, the β₁AR C-tail did not make β₂AR capable of recruiting Gα$_i$ with carvedilol stimulation. Data represent the mean ± SEM for n independent experiments as marked on the figure. Statistical significance vs. unstimulated cells was assessed using one-way ANOVA with Bonferroni correction

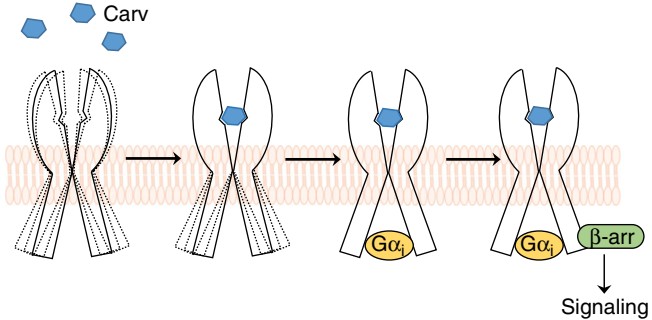

**Fig. 9** Schematic model of the carvedilol-induced $G\alpha_i$-β-arrestin-biased signaling of $\beta_1ARs$. Binding of carvedilol to the orthosteric site of the $\beta_1AR$ stablizes a distinct intermediate conformation that then promotes the recruitment of $G\alpha_i$. The carvedilol- and $G\alpha_i$-bound receptor in turn stabilizes a unique $\beta_1AR$ conformation that mediates β-arrestin-biased signaling

then removed from the system and left ventricle was excised and snap frozen in liquid nitrogen. Animal experiments carried out for this study were handled according to approved protocols and animal welfare regulations the Animal Care and Use Committee of Duke University Medical Center.

**Fluorescence resonance energy transfer measurement**. FRET measurement was performed as previously described[26]. Briefly, HEK293 cells stably expressing $\beta_1AR$-FRET sensor were cultured in glass-bottomed confocal dish. Cells were pretreated with vehicle or 200 ng per ml PTX for 16 h before experiment. On the day of experiment, cells were maintained in FRET buffer (10 mM HEPES, 0.2% BSA, 140 mM NaCl, 4.5 mM KCl, 2 mM $CaCl_2$, 2 mM $MgCl_2$, pH 7.4). FRET experiments were preformed using an Olympus IX-71 microscope. FRET was monitored as the emission ratio of YFP to Cerulean. Images were taken at 10 s interval and analyzed with ImageJ software.

**In situ proximity ligation assay**. $\beta_1AR$ or $\beta_2AR$ stable cells were cultured in 35 mm poly-D-lysine coated glass-bottom confocal dish (MatTek). Following stimulation, cells were fixed in 4% paraformaldehyde for 15 min and permeabilized with 0.2% Triton-X-100 for 10 min. After blocked with blocking buffer from Duolink Detection Kit (Sigma) at 37 °C for 30 min, cells were incubated overnight at 4 °C with anti-$\beta_1AR$ (or $\beta_2AR$) antibody from rabbit (Santa Cruz) in conjunction with anti-$G\alpha_i$ antibody from mouse (New East Biosciences). The proximity ligation reaction was performed according to the manufacturer's protocol using the Duo-link Detection Kit (Sigma). Cells were mounted with DAPI Fluoromount-G (Southern Biotech). Images were recorded with Zeiss Axio Observer Z1 confocal microscope with ×40 objective. Data analysis was performed with ImageJ software. To quantify the mean PLA signal per cell, the red PLA fluorescence intensity was divided by the number of cells. The mean PLA signal of each data set was corrected by subtracting the background staining determined as the mean PLA signal of HEK293 cells without receptor overexpression. The relative fold over non-stimulation was normalized to the mean PLA signal of the unstimulated cells. In each experiment, 20–40 cells from three images were quantified for each condition.

**EGFR internalization assessed by confocal microscopy**. HEK293 cells stably expressing FLAG-tagged $\beta_1AR$ were transfected with EGFR-GFP together with control siRNA or β-arrestin siRNA as described below. After 24 h, the transfected cells were plated into glass-bottomed confocal dish and kept in culture for additional 24 h. Following pretreatment with PTX and stimulation with ligands, cells were washed with ice-cold PBS and fixed with 4% paraformaldehyde for 15 min. EGFR internalization was visualized with Zeiss Axio Observer Z1 confocal microscope with ×63 objective. In each experiment, 50 cells of each condition were counted under microscope. The percentage of cells showing EGFR internalization was determined by the number of cells showing the intracellular aggregate of EGFR-GFP.

**EGFR internalization assessed by flow cytometry**. HEK293 cells (wildtype, β-arrestin1/2 knockout or $G\alpha_i$ knockout) were transfected with CFP-tagged $\beta_1AR$. 48 h after transfection, cells were serum starved for 4 h before ligand stimulation. Following stimulation, cells were dissociated with accutase, washed with PBS and fixed in 4% formaldehyde for 15 min at room temperature. Fixed cells were enumerated, washed twice with staining buffer (PBS, 0.5% BSA, 2 mM EDTA) and blocked with 5% rat serum (Sigma) in staining buffer for 15 min. $1 \times 10^6$ cells for each sample were stained with equal concentrations of either PE-conjugated EGFR antibody (R&D systems) or isotype control (R&D systems; PE-conjugated rat IgG2A) for 30 min at room temperature. Following staining, cells were washed twice with staining buffer and resuspended in PBS for analysis utilizing a BD LSRII flow cytometer (BD Biosciences). Data analysis was performed with FlowJo

software. Following doublet exclusion, single cells were gated for CFP positivity. To quantify relative EGFR internalization following ligand stimulation, the following formula was utilized: geometric mean fluorescence intensity of the PE-EGFR signal for each data set minus MFI of the isotype control. The resultant value was normalized to the MFI of the unstimulated cells to assess the relative percentage of EGFR internalization.

**β-arrestin siRNA knockdown**. SiRNAs targeting β-arrestin have been described previously[13]. A nonsilencing RNA duplex (5′-AAUUCUCCGAACGUGU-CACGU-3′) was used as a control. HEK293 cells stably expressing FLAG-tagged $\beta_1AR$ were seeded into 10 cm dish on the day before to reach 30–40% confluence at the time of transfection. SiRNA were transfected using GeneSilencer Transfection Reagent (Genlantis) according to the manufacturer's protocol. In brief, 20 μg siRNA and 240 μl siRNA dilution buffer were added into 180 μl serum-free medium, whereas 51 μl of transfection reagent was mixed with 300 μl serum-free medium. Both solutions were allowed to stand for 5 min at room temperature, then combined and incubated for additional 20 min. The mixture was then added to cells in the 10 cm dish with 4 ml serum-free medium. After 4 h incubation at 37 °C and 5% $CO_2$, 5.5 ml of MEM containing 20% FBS and 2% penicillin–streptomycin were added into the dish. All assays were performed 3 d after siRNA transfection.

**Statistical analysis**. Data are expressed as mean ± SEM. Statistical comparisons were performed using two-tailed Student's t-test or ANOVA with Bonferroni correction for multiple comparisons in Graphpad Prism. Normality test was performed with Shapiro-Wilk test. Outlier data points more than two standard deviations from the mean were excluded from analysis. Differences were considered statistically significant at $P < 0.05$.

**Data availability**. All data supporting the findings of this study are available from the authors upon request.

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

## Acknowledgements

We thank Dr K. Xiao (University of Pittsburgh, USA) for intellectual input. We thank Dr R.J. Lefkowitz (Duke University, USA) for providing the β-arrestin antibodies. We thank Dr M. Bouvier (University of Montreal, Canada) and Dr J.S. Gutkind (University of California, San Diego, USA) for providing the T8-βARKct construct. This work was supported by National Institutes of Health grants HL056687 and HL075443 to H.A.R. and AHA Predoctoral Fellowship 14PRE20480352 to J.W.

## Author contributions

H.A.R. supervised the entire study; H.A.R., J.W. and D.P.S. wrote the manuscript with comments from all co-authors; J.W. designed and performed most of the experiments and analyzed data; K.H. performed the ERK phosphorylation assay in mice heart; M.A.M. assisted with flow cytometry assay; G.R.D. generated the Gαi and β-arrestin knockout cells; Q.C. assisted with co-immunoprecipitation experiments; A.A. and S.E. provided material and performed preliminary experiments for β1AR FRET-sensor.

## Additional information

**Competing interests:** The authors declare no competing financial interests.

