## [Peer Review File · Nature Communications]

Reviewers' comments:

Reviewer #1 (Remarks to the Author):

In this study, the authors show that carvedilol, a ligand classically known to act as a β AR antagonist, switches the classical $G_{\alpha s}$ -coupled receptor β 1AR to a $G_{\alpha i}$ -coupled receptor, in a manner that differs from what is known for β 2AR (e.g. PKA-mediated switch). As suggested by the authors, β 1AR-mediated $G_{\alpha i}$ -coupling directs β -arrestin-dependent signaling to MAPK via EGFR, potentially through an allosteric mode of action of $G_{\alpha i}$ on the receptor/ β -arrestin complex, and which involves receptor phosphorylation—although these two last modes of regulation are not shown here. The finding about carvedilol acting as a biased ligand for promoting β 1AR- $G_{\alpha i}$ activation is compelling. The β 1AR- $G_{\alpha i}$ -dependent- β -arrestin-biased signaling is provocative, and could indeed change our view on how β -arrestin-biased signaling operates in some cases, which as suggested by the authors, would also require the biased activation of specific G proteins for some GPCRs. That being said, a major shortcoming here is the inability to directly involve β -arrestin in this process. Demonstrating the recruitment of β -arrestin to carvedilol-bound β 1AR; and/or the existence of a $G_{\alpha i}$ / β -arrestin signaling complex at the β 1AR, as well as an allosteric regulation of the receptor, would indeed validate their conclusions and model, and greatly increase the breadth of this study. As of now, it is still unclear mechanistically how $G_{\alpha i}$ binding to carvedilol-activated β 1AR regulates β -arrestin signaling, and if indeed such signaling mechanism exclusively operates under β 1AR in a conformationally-dependent manner, and if other cross-talk signaling events are involved.

Specifics:

1) It is surprising that the authors couldn't detect the binding of β -arrestin to carvedilol-activated β 1AR, since signaling through this pathway—contrarily to G protein signaling, which is an amplified response—would require the recruitment of larger amounts of this adaptor protein to the receptor, almost to stoichiometric levels of expression of β 1AR. Can the author exclude the possibility that β -arrestins are in fact recruited to EGFR instead of β 1AR to mediate MAPK activation, similarly to what was shown for other GPCRs and RTKs (Proc Natl Acad Sci U S A. 2012 24;109(17):E1028-37)? In other words, can the authors exclude the possibility of signal cross-talk rather than a functional conformational-dependent biased mechanism at the β 1AR?

2) The authors use a β 1AR FRET sensor to monitor changes in the receptor's conformation, and

show that PTX reduces—but not totally prevents—carvedilol-induced FRET signals. Are these FRET changes in PTX conditions, which would also presumably result from a change in conformation of the receptor from G α i interaction, also result from the binding of carvedilol itself, and/or the binding of other G protein subtypes or β -arrestin, for instance? It would be important for validating their model that the authors find a way to study to what extent β -arrestins participate in this conformational change (e.g. binding to the β 1AR or else). For instance, how depleting β -arrestins affect β 1AR FRET response to carvedilol. Valuable information could be gained about time frames for G α i and β -arrestin interactions with β 1AR, and perhaps support their model as whether a complex containing both G α i and β -arrestin concurrently exists under the receptor at the PM or elsewhere in the cell.

3) Residual ERK activation in PTX, β -arrestin-depleted cells, is still observed (Fig. 3a, blots). Why? Are other G α protein subtypes also recruited to carvedilol-bound β 1AR? It would be important that the authors confirm that only G α i are involved in β 1AR β -arrestin biased signaling response. In that respect, the authors show that G α i3 is activated by the carvedilol-bound β 1AR (Fig.3). What about G α i1,2, since PTX would also presumably inhibit those? Are they equally important?

4) Figs. 5 and 6; what is the efficiency of β -arrestin knockdown in cells? Is there differences in the response between β -arrestin 1 and 2?

5) The author mentioned that since PTX pre-treatment and β -arrestin knockdown did not further impair carvedilol-induced ERK activation that most likely G α i and β -arrestin are involved in the same signaling cascade, rather than acting in parallel. However, because of the lack of sensitivity of many assays used here (e.g. WB, IP, IF), which are semi-quantitative at best, the author cannot unequivocally conclude such thing (e.g. lack of additive effects). More robust and quantitative analysis of G α i and β -arrestin activity could confirm or invalidate such conclusion. In fact, a deficiency of many studies investigating biased signaling often results from the use of assays showing different levels of sensitivity, which often confound the interpretation of results.

6) The authors suggest that because the β 1AR phosphorylation by GRK and β -arrestin recruitment to the receptor are not required for G α i recruitment, that G α i acts upstream of β -arrestin; a conclusion which I do not necessarily totally concur with. Indeed, as shown recently, a complex of G protein/ β -arrestin was shown to concomitantly exist under receptors (Cell. 2016 Aug 11;166(4):907-19). Again a more careful analysis of the dynamic interactions of G α i and β -arrestin to β 1AR could clarify this. In that respect, the authors should also show the site-specific phosphorylation of distinct amino acid residues in the C-tail of the β 1AR promoted by carvedilol, if indeed this is the mechanism by which β -arrestin is recruited and distinctly activated by carvedilol (e.g. should be different from other ligands).

Reviewer #3 (Remarks to the Author):

The β_1 adrenergic receptor (β_1 AR) is recognized as a classical $G_{\alpha s}$ -coupled G-protein-coupled receptor, which also signals through β -arrestins. The β -blocker carvedilol acts as a β -arrestin-biased β adrenergic receptor (β AR) ligand that activates β -arrestin pathways while having inverse agonism towards $G_{\alpha s}$ signalling. In the article by Wang et al, the authors report the identification of a new signalling mechanism whereby the $G_{\alpha i}$ protein is required for β_1 AR-mediated β -arrestins-biased signalling. The authors present results showing that carvedilol induces a unique β_1 AR conformation resulting in the recruitment of $G_{\alpha i}$. Recruitment of $G_{\alpha i}$ to the β_1 AR is not induced by the other β AR agonists or antagonists tested, and it is not required for β_2 AR/ β -arrestin-biased signalling. It is demonstrated that both $G_{\alpha i}$ activation and β -arrestin are required for ERK activation and the internalization of the EGF receptor downstream of carvedilol-stimulated β_1 AR. The authors propose but do not demonstrate that β -arrestin acts downstream of activated $G_{\alpha i}$.

General comments

It is an interesting study, clearly written. There are however a number of issues that will have to be addressed experimentally. Despite incoherence in the presentation of some results that will need to be addressed, and missing technical details (detailed in the specific comments), the first part of the study demonstrating that carvedilol induces a specific switch in the β_1 AR conformation recruiting and activating $G_{\alpha i}$, is convincing and the data on the whole support the authors' conclusion. There are now multiple examples of specific GPCR conformations induced by distinct ligands (references in the manuscript) and a $G_{\alpha s}$ to $G_{\alpha i}$ switch has been described for β_2 AR (by a different mechanism involving $G_{\alpha s}$ and PKA) but these results concerning the β_1 AR are timely and they will be of interest to the GPCR field.

The major conceptual issue of this work is the regulation of β -arrestin signalling. The authors demonstrate that both $G_{\alpha i}$ and β -arrestin are required for carvedilol-induced responses (ERK activation and EGFR internalization) and that Pertussis Toxin pretreatment combined together with β -arrestin knock-down does not further impair ERK activation and EGFR internalization. Because they also demonstrate that neither $G_{\beta\gamma}$ nor consensus GRK or PKA-mediated receptor phosphorylation are required for carvedilol-induced $G_{\alpha i}$ recruitment to β_1 AR, the authors conclude that it is "highly likely" that $G_{\alpha i}$ is upstream of β -arrestin. They propose that the previously defined G protein bias versus β -arrestin-bias may be attributed to ligand-induced selective coupling of receptors to specific G protein subtypes, i.e G protein subtype bias, whereby a carvedilol-induced β_1 AR unique conformation recruits and activates $G_{\alpha i}$, allowing for subtype-specific GRK phosphorylation of distinct amino acids residues on the C-tail of β_1 AR, leading to the activation of downstream β -arrestin-mediated signalling.

It is an exciting hypothesis but it needs to be demonstrated and several other issues have to be addressed experimentally.

- The authors have no experimental data showing a direct molecular or functional link between G α i and β -arrestins. The demonstration of β -arrestin recruitment to the receptor upon carvedilol stimulation should be provided and the effect of PTX on this recruitment should be shown to support the proposed model. It is a critical point because other mechanisms could trigger β -arrestin signalling upon carvedilol stimulation of the β 1AR in the absence of recruitment to the receptor.

The authors state in the discussion that they have tried unsuccessfully to detect experimentally β -arrestin recruitment to the β 1AR using co-immunoprecipitation, confocal or bioluminescence resonance energy transfer –based assays. This may also be tried using other types of approaches. Indeed, successful β 1AR- β -arrestin co-immunoprecipitation has been performed in the β 1AR stable cells-line by this group, using cross-linkers (Kim et al, PNAS 2008, reference 10 in the manuscript). It is an approach that the authors could then easily use again to address the question of the recruitment of β -arrestins to the β 1AR. Other approaches can also be used to demonstrate effects on β -arrestins such as double brilliance β -arrestin conformational Biosensors (RLuc/YFP, RLucII/GFP2), which have been reported. The effect of PTX could also be tested using these approaches. Transfection of constitutively active G α i has been used as a positive control in Fig 3, it could also be used to test for its effects on β -arrestin recruitment to the β 1AR and with the double brilliance β -arrestin biosensor. These experiments should be performed to provide evidence for a direct molecular link.

- Despite some discrepancies between the individual blots and the graph reporting their quantifications in several experiments (Supplementary Figure 1, Figure 6 detailed below), the results show non-negligible remaining carvedilol-induced ERK activation (Figure 1, supplementary Figure 1 and Figure 6) and EGFR internalization (Figures 5) in β 1AR stable cells preincubated with PTX or after β -arrestin knock-down (individually or together). Although the extent of β -arrestin extinction by the siRNA is not shown (it should be added), these results suggest that part of the carvedilol-induced response is independent of G α i and potentially β -arrestin too. Because other results show that G α s/PKA signalling is not mediating G α i recruitment, how is the remaining carvedilol-induced response regulated? The authors should also address this question experimentally and/or in the discussion.

- The authors previously reported that β 1AR phosphorylation at consensus GRK sites and β -arrestin recruitment is required for EGFR transactivation in response to carvedilol (Kim et al, PNAS 2008, reference 10 in the manuscript). Here the β 1AR-GRK mutant shows no effect on ERK activation by carvedilol suggesting as stated that the phosphorylation of the β 1AR at consensus GRK sites is not important for ERK activation. How do the authors reconcile these results? Are there distinct mechanisms regulating G α i recruitment and β -arrestin recruitment/activation in response to carvedilol/ β 1AR? Are there distinct pathways regulating EGFR transactivation, EGFR internalization and ERK activation in response to carvedilol stimulation of β 1AR?

Specific comments.

The authors used FRET, western blotting, Proximity Ligand assay and confocal microscopy in their study. There are major discrepancies between the results shown on a single representative western blot membrane and the graph showing the quantification of the results from several independent experiments. It is the case for most panels/Figures showing western blot. This lack of coherence impacts negatively the comprehension and interpretation of the results. It is detailed below and should be corrected. Also there is a lack of information regarding protocols and quantification of other experiments that is detailed below. It should also be corrected.

- Supplementary Figure 1.

The graphs/quantifications do not match at all the blots shown.

. Left panel (β 1AR stable cells) : the blot from the left panel shows a maximum Carvedilol stimulation of P-ERK at 1-5 min followed by a dramatic decrease in ERK activation at 15 min, further decreased at 30-60 min. However the graph indicates maximal stimulation at 5 min, sustained ERK activation at 15 min before stabilizing at 30 and 60 min at the level observed for the 1 min time point. The blot and the graph do not match and it is particularly striking for the 1 and 15 min time points.

. Right panel (β 2AR stable cells): -blot: P-ERK stimulation by Carvedilol appears to be similar after 1 and 5 min. While PTX preincubation appears to have no effect on ERK stimulation by Carvedilol at 1 min, there is a strong decrease in P-ERK levels after 5 min Carvedilol stimulation time point. However, the graph shows no difference between cells preincubated or not with PTX after 5 min of stimulation with Carvedilol. In addition to being incoherent, the blot suggest that PTX may have an effect on ERK stimulation downstream of the β 2AR stimulated by Carvedilol possibly implicating G α i. What is representative of the overall results, the blot or the graph? They should match.

- Figure 3.

. Panel a-b: a very low magnification is shown and it is not possible to see any details. What is the magnification? Is the signal solely located at the membrane? Are there particular spots? The authors should show a zoom/ higher magnification of these pictures in parallel to the low magnification pictures and scale bars should also be included. What has been quantified (spot or intensities or both)? How was the quantification performed? The material and Methods section only indicates the use of ImageJ..... What were the Macros used? This should be corrected. How many cells have been scored in each independent experiment? The information should be provided.

. Panel c: the left Isoproterenol blot does not match the graph for the 10^{-6} M concentration. Why are the bands of Flag (β 2AR) blots of the lysates from the β 2AR stable cells are so diffused compared to the ones of the β 1AR? It is due to glycosylation? It is an important issue because it is used for the quantification. Is it reliable? What is representative of the results, the blot or the

graph? They should match.

. Panel d: the graph does not match the blot. On the blot, Isoproterenol stimulation of β 1AR stable cells results in the same levels of activated $G_{\alpha i}$ than the one observed for the 30 min stimulation with Carvedilol, while the graph shows no activated $G_{\alpha i}$ in response to Isoproterenol. What is representative of the results, the blot or the graph? They should match.

The authors should also do this experiment again and perform a Co-IP similar to the one shown in panel c but blot for activated $G_{\alpha i}$ instead of total $G_{\alpha i}$. This may provide additional interesting information.

- Figure 4.

. Panel a: the same modifications as indicated above for panels 3a-b should be performed and the additional missing information should also be provided

. Panel b: the Carvedilol control is missing and the blots should be shown in a supplementary figure.

- Figure 5.

. The same modifications as indicated above for panels 3a-b should be performed for both panels and the additional missing information should also be provided.

. The authors should show the extent of siRNA-mediated β -arrestins extinction (see also below).

. The authors should definitely also perform FACS analysis to confirm by another approach the results shown in panel a and b. It is an easy experiment to perform and results can be readily quantified to demonstrate the effects of both PTX and β -arrestins extinction (together and individually).

- Figure 6.

. Panel a: the graph does not match the blot. It is a critical experiment of the manuscript.

The blot shows a small decrease in P-ERK activation in response to 30 min Carvedilol stimulation compared to the 5 min time point. This decrease is not apparent in the graph. The blot also shows that PTX preincubation reduces the level of P-ERK observed after 5 min Carvedilol stimulation to a level similar to the one observed after 30 min stimulation in absence of PTX. The graph shows that PTX abolishes P-ERK stimulation in response to 5min Carvedilol stimulation. To make it short, the blot shows that there is still some non-negligible ERK activation after 5min Carvedilol stimulation when the cells have been preincubated with PTX and the graph shows the opposite. The same comment can be made for the effect of siRNA-mediated β -arrestins. What is representative of the results, the blot or the graph? They should match.

This is a critical issue since the authors have not demonstrated a link between $G_{\alpha i}$ recruitment/activation and β -arrestins (see introductory comments).

The authors should also show the extent of siRNA-mediated β -arrestins extinction (see also below).

. Panel c: same comment as above for panel a. The blot P-ERK levels +/- PTX do not match the quantification shown by the graph.

-Figure 7

. Panel a: the quality of the blots is really poor while using the same approach (transient transfection) in panel c, the blots are of good quality. Better blots should be shown.

What is the level of P-ERK in this experiment? It should be shown.

. Panel c: what is the level of P-ERK in the left panel experiments? Is ERK activated in the absence of G α i recruitment by the β 1AR chimera containing the β 2AR C-tail?

Minor comments.

- lane 37: “binging” should be replaced by “binding”

- lane 43: “... unlike that previously for any known G...” should be corrected

- Legend Figure 4: “(b) β 1AR or β 1AR stable cells....” Should be: “(b) β 1AR or β 2AR stable cells....”

We thank the reviewers for their time and helpful suggestions to improve our manuscript. We have performed most of the requested experiments and many additional experiments that strengthen the findings within. Please find our point-by-point responses below in blue and the changes within the manuscript in red.

Reviewers comments:

Reviewer #1:

In this study, the authors show that carvedilol, a ligand classically known to act as a β AR antagonist, switches the classical $G\alpha_s$ -coupled receptor β_1 AR to a $G\alpha_i$ -coupled receptor, in a manner that differs from what is known for β_2 AR (e.g. PKA-mediated switch). As suggested by the authors, β_1 AR-mediated $G\alpha_i$ -coupling directs β -arrestin-dependent signaling to MAPK via EGFR, potentially through an allosteric mode of action of $G\alpha_i$ on the receptor/ β -arrestin complex, and which involves receptor phosphorylation—although these two last modes of regulation are not shown here. The finding about carvedilol acting as a biased ligand for promoting β_1 AR- $G\alpha_i$ activation is compelling. The β_1 AR- $G\alpha_i$ -dependent- β -arrestin-biased signaling is provocative, and could indeed change our view on how β -arrestin-biased signaling operates in some cases, which as suggested by the authors, would also require the biased activation of specific G proteins for some GPCRs. That being said, a major shortcoming here is the inability to directly involve β -arrestin in this process. Demonstrating the recruitment of β -arrestin to carvedilol-bound β_1 AR; and/or the existence of a $G\alpha_i$ / β -arrestin signaling complex at the β_1 AR, as well as an allosteric regulation of the receptor, would indeed validate their conclusions and model, and greatly increase the breadth of this study. As of now, it is still unclear mechanistically how $G\alpha_i$ binding to carvedilol-activated β_1 AR regulates β -arrestin signaling, and if indeed such signaling mechanism exclusively operates under β_1 AR in a conformationally-dependent manner, and if other cross-talk signaling events are involved.

Specifics:

1) It is surprising that the authors couldn't detect the binding of β -arrestin to carvedilol-activated β_1 AR, since signaling through this pathway—contrarily to G protein signaling, which is an amplified response—would require the recruitment of larger amounts of this adaptor protein to the receptor, almost to stoichiometric levels of expression of β_1 AR. Can the author exclude the possibility that β -arrestins are in fact recruited to EGFR instead of β_1 AR to mediate MAPK activation, similarly to what was shown for other GPCRs and RTKs (Proc Natl Acad Sci U S A. 2012 24;109(17):E1028-37)? In other words, can the authors exclude the possibility of signal cross-talk rather than a functional conformational-dependent biased mechanism at the β_1 AR?

We have performed many additional experiments to explore the recruitment of β -arrestin to the carvedilol-stimulated β_1 AR (Reviewer Figure 1) and described below. Unfortunately despite our nearly exhaustive efforts, we have been unable to detect recruitment of β -arrestin to carvedilol-stimulated β_1 ARs in a number of assays.

(1) In the co-immunoprecipitation experiments (Reviewer Figure 1a, representative blots of five independent experiments), we show that the balanced agonist isoproterenol significantly increased the amount of β -arrestin bound to β_1 ARs, which was not observed with carvedilol stimulation. While we acknowledge that in our previous study we were

able to show carvedilol-induced β -arrestin recruitment to β_1 ARs (Kim et al, PNAS 2008), at this time we are unable to demonstrate recruitment by co-IP.

- (2) We tried overexpressing β -arrestins (Reviewer Figure 1b, representative blots of four independent experiments), in addition to several GRKs (Reviewer Figure 1c, representative blots of four independent experiments). In all the conditions, β -arrestin recruitment was seen with isoproterenol stimulation, but not with carvedilol stimulation.
- (3) In a panel of FRET sensors (Reviewer Figure 1d), the C-terminus of β_1 AR is tagged with Cerulean, the N-terminus or C-terminus of β -arrestin1 or β -arrestin2 is tagged with YFP, and FRET between the two was monitored. Stimulation with norepinephrine (NE) increased the FRET ratio, indicating the NE-induced β -arrestin recruitment to the receptor, whereas carvedilol did not have significant effect (Reviewer Figure 1d).
- (4) As binding of β -arrestin to β_1 AR is required for subsequent receptor desensitization and internalization via clathrin-coated pits, we also monitored the translocation of β -arrestin to the plasma membrane with total internal reflection fluorescence (TIRF) microscopy (Reviewer Figure 1e), which allows the selective visualization of the plasma membrane and the cytoplasmic region immediately beneath the plasma membrane. Upon isoproterenol stimulation, the bright puncta of the GFP-tagged β -arrestin2 was observed, and the amount of β -arrestin2 was increased, indicating the recruitment of the β -arrestins to the membrane-localized receptors. In contrast, the localization and the amount of GFP-tagged β -arrestin2 was not affected by carvedilol.
- (5) β -arrestin binding to the activated receptor induces internalization of the receptor/ β -arrestin complex to endosomes. In the DiscoverX Pathhunter assay (Reviewer Figure 1f), the endosome and β -arrestin are tagged with two fragments of the β -galactosidase enzyme, thus allowing the detection of β -arrestin translocation to endosomes by the increase of chemiluminescent. Isoproterenol dose-dependently increased the signal, while carvedilol did not have significant effect (Reviewer Figure 1f).
- (6) As suggested by Reviewer 2, we monitored β -arrestin activation by the BRET-based double brilliance β -arrestin conformational sensor, in which the N-terminus and C-terminus of β -arrestin2 are tagged with Rluc and YFP, respectively (Reviewer Figure 1g). The BRET ratio was increased with isoproterenol stimulation, whereas it remained unaltered with carvedilol stimulation.

We believe our inability to detect carvedilol stimulated β -arrestin translocation is most likely because as a class A receptor the interaction of the β_1 AR with β -arrestin is transient and of low affinity. In discussion with Drs. Lefkowitz and Caron, both who have extensive experience studying β -arrestin recruitment to the β_2 AR, carvedilol-induced β -arrestin recruitment would only be observed if the C-terminus of the β_1 AR were exchanged with the class B vasopressin type 2 receptor (V_2 R) as they have shown using the β_2 AR- V_2 R chimera. Unfortunately, such a strategy of swapping the V_2 R c-tail with the β_1 AR c-tail is not a viable option in our study since as we show in our paper (Figure 7), the c-tail of the β_1 AR is essential for $G\alpha_i$ binding and signaling.

Though we do not have direct evidence for the recruitment of β -arrestin to the carvedilol-stimulated β_1 ARs, we indirectly addressed this question by studying of the importance of β -arrestin in stabilizing a β_1 AR conformation induced by carvedilol. In the FRET assay with the

β_1 AR conformation sensor, the carvedilol-induced FRET change was diminished in cells lacking β -arrestin (Reviewer Figure 2a), suggesting that β -arrestin is recruited to the carvedilol-bound β_1 AR and is involved in allosterically stabilizing this receptor conformation.

Thank you for the suggestion on testing the recruitment of β -arrestins to EGFRs. In β_1 AR stable cells transfected with GFP-tagged EGFRs and HA-tagged β -arrestins, we performed co-immunoprecipitation assay of the EGFR and β -arrestins after stimulation with isoproterenol, carvedilol or EGF. Each of the ligands activated the ERK phosphorylation, but did not induce binding of β -arrestin to the EGF receptor, indicating that its recruitment is not required for downstream EGFR signaling (Reviewer Figure 2b, representative blots of three independent experiments).

2) The authors use a β_1 AR FRET sensor to monitor changes in the receptor's conformation, and show that PTX reduces—but not totally prevents—carvedilol-induced FRET signals. Are these FRET changes in PTX conditions, which would also presumably result from a change in conformation of the receptor from $G\alpha_i$ interaction, also result from the binding of carvedilol itself, and/or the binding of other G protein subtypes or β -arrestin, for instance? It would be important for validating their model that the authors find a way to study to what extent β -arrestins participate in this conformational change (e.g. binding to the β_1 AR or else). For instance, how depleting β -arrestins affect β_1 AR FRET response to carvedilol. Valuable information could be gained about time frames for $G\alpha_i$ and β -arrestin interactions with β_1 AR, and perhaps support their model as whether a complex containing both $G\alpha_i$ and β -arrestin concurrently exists under the receptor at the PM or elsewhere in the cell.

These are very good suggestions to strengthen our manuscript. We performed carvedilol competition binding experiments, and showed that pretreatment with PTX does not alter the affinity of carvedilol to the receptors (Reviewer Figure 2c). These data demonstrate that the effect of PTX on the FRET change does not result from a change in affinity of the β_1 AR for carvedilol. As described above, we also tested the effect of β -arrestin depletion on the receptor conformational change using the β_1 AR FRET sensor, and showed that the carvedilol-induced FRET change is diminished in the β -arrestin knockout cells (Reviewer Figure 2a). Taking together, our results are consistent with the concept that both $G\alpha_i$ and β -arrestins are involved in the carvedilol-activated β_1 AR conformational state.

3) Residual ERK activation in PTX, β -arrestin-depleted cells, is still observed (Fig. 3a, blots). Why? Are other $G\alpha$ protein subtypes also recruited to carvedilol-bound β_1 AR? It would be important that the authors confirm that only $G\alpha_i$ are involved in β_1 AR β -arrestin biased signaling response. In that respect, the authors show that $G\alpha_{i3}$ is activated by the carvedilol-bound β_1 AR (Fig.3). What about $G\alpha_{i1,2}$, since PTX would also presumably inhibit those? Are they equally important?

We suspected that the residual ERK activation was likely due to incomplete blockade of $G\alpha_i$ by PTX or the incomplete knockdown of β -arrestins with siRNA. Since the experiments were performed in β_1 AR stable cells with high expression level of the receptors, it seemed possible that remaining $G\alpha_i$ activity or β -arrestin levels were sufficient to mediate some degree of downstream ERK signaling. This possibility was also supported by our *in vivo* experiments, whereby in the β_2 AR knockout hearts with endogenous expressed β_1 AR levels, PTX completely blocked carvedilol-induced ERK activation.

To test this hypothesis, we generated new cell lines that are completely deficient of $G\alpha_i$ or β -arrestin using CRISPR/Cas9 gene editing. In $G\alpha_i$ knockout cells that we generated, all three subtypes were depleted with CRISPR gene editing. In β_1 AR stable cells lacking $G\alpha_i$, the carvedilol-induced ERK activation was completely blocked (new Figure 1b). In contrast, $G\alpha_i$ knockout in β_2 AR stable cells showed robust carvedilol-stimulated ERK activation (new Figure 1b). Moreover, the knockout of either $G\alpha_i$ or β -arrestin1/2 completely blocked the carvedilol-induced β_1 AR-mediated ERK activation (new Figure 6b). These data are consistent with our earlier PTX and siRNA experiments and now more robustly show that both $G\alpha_i$ and β -arrestin are required for carvedilol-stimulated β_1 AR signaling.

The three subtypes of $G\alpha_i$ share high sequence similarity, and were not differentiated in most assays in this study. In Figure 3a and 3c, the antibodies used recognize all three $G\alpha_i$ subtypes. In Figure 3b, the recruitment of $G\alpha_{i3}$ was shown as a representative. We performed additional experiments and now show the recruitment of both $G\alpha_{i1}$ and of $G\alpha_{i2}$ (new Supplementary Figure 3a).

4) Figs. 5 and 6; what is the efficiency of β -arrestin knockdown in cells? Is there a difference in the response between β -arrestin1 and 2?

The western blots showing the efficiency of β -arrestin knockdown are now included in the figures (Figure 6a and new Supplementary Figure 3a). To more rigorously determine whether β -arrestin1 and 2 have distinct roles in the signaling, we tested the ERK activation in newly generated β -arrestin1 and β -arrestin2 knockout cell lines (new Figure 6b). These data show that depleting either β -arrestin1 or β -arrestin2 significantly diminishes carvedilol-induced β_1 AR-mediated ERK activation, indicating that both β -arrestins are required for carvedilol-stimulated signaling.

5) The author mentioned that since PTX pre-treatment and β -arrestin knockdown did not further impair carvedilol-induced ERK activation that most likely $G\alpha_i$ and β -arrestin are involved in the same signaling cascade, rather than acting in parallel. However, because of the lack of sensitivity of many assays used here (e.g. WB, IP, IF), which are semi-quantitative at best, the author cannot unequivocally conclude such a thing (e.g. lack of additive effects). More robust and quantitative analysis of $G\alpha_i$ and β -arrestin activity could confirm or invalidate such a conclusion. In fact, a deficiency of many studies investigating biased signaling often results from the use of assays showing different levels of sensitivity, which often confound the interpretation of results.

Thank you for these insightful comments and suggestions. We believe that these data using our newly generated $G\alpha_i$ and β -arrestin1/2 knockout cell lines more robustly demonstrate the dual requirement for both these transducers for this signaling pathway. Since ERK activation is an amplified downstream signaling system, these new data showing complete blockade by either $G\alpha_i$ or β -arrestins knockout suggests that it is unlikely that $G\alpha_i$ and β -arrestin act independently in the signaling cascade (new Figure 1b and 6b). We agree with the reviewer that more sensitive and quantitative assays could be helpful in interpreting the results and the support of conclusion. Unfortunately, in a number of assays we tried to directly detect the β -arrestin translocation or activation, as mentioned in Point 1, we were unable to detect carvedilol-stimulated β -arrestin translocation to wild-type β_1 ARs. Therefore, we used the known β -arrestin-dependent signaling, the ERK activation and EGFR transactivation, as signaling readouts in this study. As requested we performed new experiments using flow cytometry as a

more quantitative assay to monitor the EGFR internalization (new Figure 5c). These data confirm our earlier confocal data and are now included in the paper.

6) The authors suggest that because the β_1 AR phosphorylation by GRK and β -arrestin recruitment to the receptor are not required for $G\alpha_i$ recruitment, that $G\alpha_i$ acts upstream of β -arrestin; a conclusion which I do not necessarily totally concur with. Indeed, as shown recently, a complex of G protein/ β -arrestin was shown to concomitantly exist under receptors (Cell. 2016 Aug 11;166(4):907-19). Again a more careful analysis of the dynamic interactions of $G\alpha_i$ and β -arrestin to β_1 AR could clarify this. In that respect, the authors should also show the site-specific phosphorylation of distinct amino acid residues in the C-tail of the β_1 AR promoted by carvedilol, if indeed this is the mechanism by which β -arrestin is recruited and distinctly activated by carvedilol (e.g. should be different from other ligands).

As discussed above in Point 5, our new data showing the complete blockade of ERK activation by either $G\alpha_i$ or β -arrestins depletion support the concept that $G\alpha_i$ and β -arrestins are likely to act in the same signaling cascade. In the respect of the order of signaling components in this cascade, as GRK-mediated receptor phosphorylation is a prerequisite for β -arrestin recruitment and subsequent signaling, but not required for $G\alpha_i$ recruitment, we speculate that $G\alpha_i$ acts upstream to β -arrestin binding.

Ongoing work in the laboratory will determine the barcode phosphorylation pattern of the β_1 AR induced by carvedilol or by isoproterenol. Given the extensive nature of this work we believe it is beyond the scope of this study. Therefore, we have opted to remove it from the current model and discuss it as a future direction in the manuscript.

Reviewer #2:

The β_1 adrenergic receptor (β_1 AR) is recognized as a classical $G\alpha_s$ -coupled G-protein-coupled receptor, which also signals through β -arrestins. The β -blocker carvedilol acts as a β -arrestin-biased β adrenergic receptor (β AR) ligand that activates β -arrestin pathways while having inverse agonism towards $G\alpha_s$ signalling. In the article by Wang et al, the authors report the identification of a new signalling mechanism whereby the $G\alpha_i$ protein is required for β_1 AR-mediated β -arrestins-biased signalling. The authors present results showing that carvedilol induces a unique β_1 AR conformation resulting in the recruitment of $G\alpha_i$. Recruitment of $G\alpha_i$ to the β_1 AR is not induced by the other β AR agonists or antagonists tested, and it is not required for β_2 AR/ β arrestin-biased signalling. It is demonstrated that both $G\alpha_i$ activation and β -arrestin are required for ERK activation and the internalization of the EGF receptor downstream of carvedilol-stimulated β_1 AR. The authors propose but do not demonstrate that β -arrestin acts downstream of activated $G\alpha_i$.

General comments

It is an interesting study, clearly written. There are however a number of issues that will have to be addressed experimentally. Despite incoherence in the presentation of some results that will need to be addressed, and missing technical details (detailed in the specific comments), the first part of the study demonstrating that carvedilol induces a specific switch in the β_1 AR conformation recruiting and activating $G\alpha_i$, is convincing and the data on the whole support the authors' conclusion. There are now multiple examples of specific GPCR conformations induced by distinct ligands (references in the manuscript) and a $G\alpha_s$ to $G\alpha_i$ switch has been described

for β_2 AR (by a different mechanism involving $G\alpha_s$ and PKA) but these results concerning the β_1 AR are timely and they will be of interest to the GPCR field.

The major conceptual issue of this work is the regulation of β -arrestin signalling. The authors demonstrate that both $G\alpha_i$ and β -arrestin are required for carvedilol-induced responses (ERK activation and EGFR internalization) and that Pertussis Toxin pretreatment combined together with β -arrestin knock-down does not further impair ERK activation and EGFR internalization. Because they also demonstrate that neither $G\beta\gamma$ nor consensus GRK or PKA-mediated receptor phosphorylation are required for carvedilol-induced $G\alpha_i$ recruitment to β_1 AR, the authors conclude that it is “highly likely” that $G\alpha_i$ is upstream of β -arrestin. They propose that the previously defined G protein bias versus β -arrestin-bias may be attributed to ligand-induced selective coupling of receptors to specific G protein subtypes, i.e G protein subtype bias, whereby a carvedilol-induced β_1 AR unique conformation recruits and activates $G\alpha_i$, allowing for subtype-specific GRK phosphorylation of distinct amino acids residues on the C-tail of β_1 AR, leading to the activation of downstream β -arrestin-mediated signalling.

It is an exciting hypothesis but it needs to be demonstrated and several other issues have to be addressed experimentally.

- The authors have no experimental data showing a direct molecular or functional link between $G\alpha_i$ and β -arrestins. The demonstration of β -arrestin recruitment to the receptor upon carvedilol stimulation should be provided and the effect of PTX on this recruitment should be shown to support the proposed model. It is a critical point because other mechanisms could trigger β -arrestin signalling upon carvedilol stimulation of the β_1 AR in the absence of recruitment to the receptor.

The authors state in the discussion that they have tried unsuccessfully to detect experimentally β -arrestin recruitment to the β_1 AR using co-immunoprecipitation, confocal or bioluminescence resonance energy transfer –based assays. This may also be tried using other types of approaches. Indeed, successful β_1 AR- β -arrestin co-immunoprecipitation has been performed in the β_1 AR stable cells-line by this group, using cross-linkers (Kim et al, PNAS 2008, reference 10 in the manuscript). It is an approach that the authors could then easily use again to address the question of the recruitment of β -arrestins to the β_1 AR. Other approaches can also be used to demonstrate effects on β -arrestins such as double brilliance β -arrestin conformational Biosensors (RLuc/YFP, RLucII/GFP2), which have been reported. The effect of PTX could also be tested using these approaches. Transfection of constitutively active $G\alpha_i$ has been used as a positive control in Fig 3, it could also be used to test for its effects on β -arrestin recruitment to the β_1 AR and with the double brilliance β -arrestin biosensor. These experiments should be performed to provide evidence for a direct molecular link.

Thank you for the thoughtful suggestions. We have spent a significant amount of time optimizing a number of assays, including the ones suggested by the reviewer, to detect carvedilol-induced β -arrestin recruitment to the β_1 AR. Unfortunately, as we explain in detail above (Reviewer 1 Point 1), we have been completely unsuccessful. Please see our response to Point 1 of Reviewer 1 for the details.

- Despite some discrepancies between the individual blots and the graph reporting their quantifications in several experiments (Supplementary Figure 1, Figure 6 detailed below), the results show non- negligible remaining carvedilol-induced ERK activation (Figure 1, supplementary Figure 1 and Figure 6) and EGFR internalization (Figures 5) in β_1 AR stable cells preincubated with PTX or after β -arrestin knock-down (individually or together). Although the

extent of β -arrestin extinction by the siRNA is not shown (it should be added), these results suggest that part of the carvedilol-induced response is independent of $G\alpha_i$ and potentially β -arrestin too. Because other results show that $G\alpha_s$ /PKA signalling is not mediating $G\alpha_i$ recruitment, how is the remaining carvedilol-induced response regulated? The authors should also address this question experimentally and/or in the discussion.

We suspected that the residual ERK activation was likely due to incomplete blockade of $G\alpha_i$ by PTX or the incomplete knockdown of β -arrestins with siRNA. Since the experiments were performed in β_1 AR stable cells with high expression level of the receptors, it seemed possible that remaining $G\alpha_i$ activity or β -arrestin levels were sufficient to mediate some degree of downstream ERK signaling. This possibility was also supported by our *in vivo* experiments, whereby in the β_2 AR knockout hearts with endogenous expressed β_1 AR levels, PTX completely blocked carvedilol-induced ERK activation.

To more robustly test this concept, we generated new cells lines that are completely deficient of $G\alpha_i$ or β -arrestin using CRISPR/Cas9 gene editing. In $G\alpha_i$ knockout cells that we generated, all three subtypes were depleted with CRISPR gene editing. In β_1 AR stable cells lacking $G\alpha_i$, the carvedilol-induced ERK activation was completely blocked (new Figure 1b). In contrast, $G\alpha_i$ knockout in β_2 AR stable cells showed robust carvedilol stimulated ERK activation (new Figure 1b). Moreover, the knockout of either $G\alpha_i$ or β -arrestin1/2 completely blocked the carvedilol-induced β_1 AR-mediated ERK activation (new Figure 6b). These data are consistent with our earlier PTX and siRNA experiments and now more conclusively show that both $G\alpha_i$ and β -arrestin are required for carvedilol-stimulated β_1 AR signaling.

- The authors previously reported that β_1 AR phosphorylation at consensus GRK sites and β -arrestin recruitment is required for EGFR transactivation in response to carvedilol (Kim et al, PNAS 2008, reference 10 in the manuscript). Here the β_1 AR-GRK mutant shows no effect on ERK activation by carvedilol suggesting as stated that the phosphorylation of the β_1 AR at consensus GRK sites is not important for ERK activation. How do the authors reconcile these results? Are there distinct mechanisms regulating $G\alpha_i$ recruitment and β -arrestin recruitment/activation in response to carvedilol/ β_1 AR? Are there distinct pathways regulating EGFR transactivation, EGFR internalization and ERK activation in response to carvedilol stimulation of β_1 AR?

We apologize for not making the statement clearer. In the original manuscript, we showed that GRK- β_1 AR mutant had no effect on $G\alpha_i$ recruitment indicating that phosphorylation of the c-tail is not required for carvedilol induced $G\alpha_i$ interaction with the receptor. In previous published work, we have shown that mutation of these GRK-phosphorylation sites on the β_1 AR prevents isoproterenol-stimulated β -arrestin recruitment (Rapacciuolo et al., JBC 2003) and carvedilol-induced EGFR transactivation (Kim et al., PNAS 2008). In this study, we confirmed these results showing the inability of the GRK- β_1 AR to activate ERK signaling in response to carvedilol (added to Figure 7a). We interpret these data to indicate that $G\alpha_i$ and β -arrestin bind to distinct intermediate conformations of the carvedilol-occupied β_1 AR.

Specific comments. The authors used FRET, western blotting, Proximity Ligand assay and confocal microscopy in their study. There are major discrepancies between the results shown on a single representative western blot membrane and the graph showing the quantification of the results from several independent experiments. It is the case for most panels/Figures showing western blot. This lack of coherence impacts negatively the comprehension and

interpretation of the results. It is detailed below and should be corrected. Also there is a lack of information regarding protocols and quantification of other experiments that is detailed below. It should also be corrected.

- Supplementary Figure 1.

The graphs/quantifications do not match at all the blots shown.

. Left panel (β_1 AR stable cells): the blot from the left panel shows a maximum Carvedilol stimulation of P-ERK at 1-5 min followed by a dramatic decrease in ERK activation at 15 min, further decreased at 30-60 min. However the graph indicates maximal stimulation at 5 min, sustained ERK activation at 15 min before stabilizing at 30 and 60 min at the level observed for the 1 min time point. The blot and the graph do not match and it is particularly striking for the 1 and 15 min time points.

. Right panel (β_2 AR stable cells): -blot: P-ERK stimulation by Carvedilol appears to be similar after 1 and 5 min. While PTX preincubation appears to have no effect on ERK stimulation by Carvedilol at 1 min, there is a strong decrease in P-ERK levels after 5 min Carvedilol stimulation time point. However, the graph shows no difference between cells preincubated or not with PTX after 5 min of stimulation with Carvedilol. In addition to being incoherent, the blot suggest that PTX may have an effect on ERK stimulation downstream of the β_2 AR stimulated by Carvedilol possibly implicating $G\alpha_i$. What is representative of the overall results, the blot or the graph? They should match.

Thanks for pointing out. In our experiments, there were some variation of the peak time and the signal level of ERK phosphorylation. The quantification graph showed the average of signal of all experiments. For the β_1 AR stable cells, we have changed to another representative blot in the revised manuscript. In the β_2 AR stable cells, there was no statistically significant difference between the control group and the PTX group at each time point. To confirm this, we performed additional experiments and included the data in new Supplementary Figure 1a. We also changed representative blots for this panel.

- Figure 3.

. Panel a-b: a very low magnification is shown and it is not possible to see any details. What is the magnification? Is the signal solely located at the membrane? Are there particular spots? The authors should show a zoom/ higher magnification of these pictures in parallel to the low magnification pictures and scale bars should also be included. What has been quantified (spot or intensities or both)? How was the quantification performed? The material and Methods section only indicates the use of ImageJ..... What were the Macros used? This should be corrected. How many cells have been scored in each independent experiment? The information should be provided.

Thanks for the suggestions. We have now included the magnified image of areas of the original pictures. We also added the detailed information into the method section.

. Panel c: the left Isoproterenol blot does not match the graph for the 10^{-6} M concentration. Why are the bands of Flag (β_2 AR) blots of the lysates from the β_2 AR stable cells are so diffused compared to the ones of the β_1 AR? It is due to glycosylation? It is an important issue because it is used for the quantification. Is it reliable? What is representative of the results, the blot or the graph? They should match.

We have changed the representative blots. The diffused band pattern of β_2 ARs is due to the glycosylation of the receptors, as β_2 ARs have three glycosylation sites (Asn6, Asn15, Asn187) (Mialet-Perez et al., JBC 2004) whereas β_1 ARs have one predicted site (Asn15) (He et al., BBRC 2002). To get a reliable quantification of the β_2 AR amount, we quantified the densitometry of the entire band.

. Panel d: the graph does not match the blot. On the blot, Isoproterenol stimulation of β_1 AR stable cells results in the same levels of activated $G\alpha_i$ than the one observed for the 30 min stimulation with Carvedilol, while the graph shows no activated $G\alpha_i$ in response to Isoproterenol. What is representative of the results, the blot or the graph? They should match. The authors should also do this experiment again and perform a Co-IP similar to the one shown in panel c but blot for activated $G\alpha_i$ instead of total $G\alpha_i$. This may provide additional interesting information.

We have changed the representative blots. We also performed the co-immunoprecipitation of FLAG- β_1 ARs and active $G\alpha_i$ (Reviewer Figure 3a). With either IP the receptors and detecting active $G\alpha_i$, or IP active $G\alpha_i$ and detecting receptor, we did not observe the binding of β_1 ARs with activated $G\alpha_i$. This is consistent with current understanding of G protein activation cycle: GDP/GTP exchange drives the dissociation of $G\alpha$ from $G\beta\gamma$ and the receptor. The dissociated GTP-bound $G\alpha$ and the $G\beta\gamma$ can interact with their specific effector proteins. The intrinsic GTPase activity of $G\alpha$ then catalyzes the GTP hydrolysis, resulting the GDP-bound $G\alpha$ which then dissociates from effector proteins and re-associates with $G\beta\gamma$.

- Figure 4.

. Panel a: the same modifications as indicated above for panels 3a-b should be performed and the additional missing information should also be provided
. Panel b: the Carvedilol control is missing and the blots should be shown in a supplementary figure.

Thank you for these suggestions. We have added magnified images in Figure 4a, and included the representative blots for quantifications in Figure 4b (new Supplementary Figure 4). The response of carvedilol is shown in Figure 3c.

- Figure 5.

. The same modifications as indicated above for panels 3a-b should be performed for both panels and the additional missing information should also be provided.
. The authors should show the extent of siRNA-mediated β -arrestins extinction (see also below).
. The authors should definitely also perform FACS analysis to confirm by another approach the results shown in panel a and b. It is an easy experiment to perform and results can be readily quantified to demonstrate the effects of both PTX and β -arrestins extinction (together and individually).

We have magnified and cropped pictures in Figure 5a and 5b for better presentation, and added details in the method section. Blots showing the extent of β -arrestins knockdown is shown in new Supplementary Figure 5a.

Thank you for the advice on assessing the EGFR internalization with FACS. This is a very good suggestion to strengthen our manuscript. For better understanding the role of β -arrestins and $G\alpha_i$ in this pathway, we utilized the β -arrestins knockout and $G\alpha_i$ knockout cells generated with

CRISPR-Cas9 gene editing (new Supplementary Figure 5b). Using flow cytometry to quantify the EGFR level on cell surface, we compared the ligand-induced EGFR internalization in wild type, β -arrestins or $G\alpha_i$ knockout cells transfected with CFP-tagged β_1 ARs. Consistent with the results of confocal microscopy assay, we found that the depletion of either β -arrestin or $G\alpha_i$ abrogated the carvedilol-induced EGFR internalization, suggesting that both transducers are required for this response. In contrast, the isoproterenol-induced response is impaired in β -arrestin knockout cells, but not in $G\alpha_i$ knockout cells.

- Figure 6.

. Panel a: the graph does not match the blot. It is a critical experiment of the manuscript. The blot shows a small decrease in P-ERK activation in response to 30 min Carvedilol stimulation compared to the 5 min time point. This decrease is not apparent in the graph. The blot also shows that PTX preincubation reduces the level of P-ERK observed after 5 min Carvedilol stimulation to a level similar to the one observed after 30 min stimulation in absence of PTX. The graph shows that PTX abolishes P-ERK stimulation in response to 5min Carvedilol stimulation. To make it short, the blot shows that there is still some non-negligible ERK activation after 5min Carvedilol stimulation when the cells have been preincubated with PTX and the graph shows the opposite. The same comment can be made for the effect of siRNA-mediated β -arrestins. What is representative of the results, the blot or the graph? They should match.

This is a critical issue since the authors have not demonstrated a link between $G\alpha_i$ recruitment/activation and β -arrestins (see introductory comments).

The authors should also show the extent of siRNA-mediated β -arrestins extinction (see also below).

. Panel c: same comment as above for panel a. The blot P-ERK levels +/- PTX do not match the quantification shown by the graph.

There were some variations of ERK phosphorylation level and the extent of PTX effect between individual experiments, and the graph is the average quantified densitometry of all experiments. As mentioned above in major comments regarding the residual ERK after PTX or β -arrestin siRNA, we suspect the high receptor level, the remaining $G\alpha_i$ activity or β -arrestins expression may be the reason for remaining ERK activation. We hope our new data with the $G\alpha_i$ or β -arrestin knockout cells (new Figure 1b and 6b) will help clarify the results, as the depletion of $G\alpha_i$ or β -arrestins with gene editing had a much more robust effect on carvedilol-induced ERK activation.

-Figure 7

. Panel a: the quality of the blots is really poor while using the same approach (transient transfection) in panel c, the blots are of good quality. Better blots should be shown. What is the level of P-ERK in this experiment? It should be shown.

. Panel c: what is the level of P-ERK in the left panel experiments? Is ERK activated in the absence of $G\alpha_i$ recruitment by the β_1 AR chimera containing the β_2 AR C-tail?

We have changed the representative blots and included the blots of ERK phosphorylation level in Figure 7a. We have tested the ERK activation mediated by the β AR chimera mutants (Reviewer Figure 3b). When stimulated with isoproterenol, both the chimera mutants, $\beta_{1/2}$ AR and $\beta_{2/1}$ AR, activated ERK to the same extent of wild type β_1 AR and β_2 AR. In contrast, the carvedilol-induced ERK activation is impaired in the $\beta_{1/2}$ AR-mediated signaling. This is

consistent with our hypothesis that $G\alpha_i$ is required for the carvedilol-induced β_1 AR signaling, as the $\beta_{1/2}$ AR was unable to recruit $G\alpha_i$ upon carvedilol stimulation.

Minor comments.

- lane 37: “binging” should be replaced by “binding”
- lane 43: “”... unlike that previously for any known G...” should be corrected
- Legend Figure 4: “(b) β_1 AR or β_1 AR stable cells...” Should be: “(b) β_1 AR or β_2 AR stable cells...”

Thank you for pointing this out and have corrected accordingly.

a. co-IP: β_1 AR stable cells

b. co-IP: β_1 AR stable cells transfected with HA- β -arrestin1 and 2

c. co-IP: β_1 AR stable cells transfected with HA- β -arrestin1/2 and GRKs

d. FRET: HEK293 cells transfected with Cerulean-tagged β_1 AR and β -arrestin1/2 with YFP at N- or C- terminus.

e. Total internal reflection fluorescence (TIRF) microscopy: β_1 AR stable cells transfected with GFP- β -arrestin2

f. DiscoverX Pathhunter assay: β_1 AR endocytosis represented by β -arrestin recruitment to endosomes

g. BRET-based double brilliance β -arrestin conformational sensor

a. Wild type or β -arrestin knockout HEK293 cells transfected with β_1 AR conformational FRET sensor

b. co-IP: β_1 AR stable cells transfected with HA- β -arrestin1/2 and GFP-EGFR

c. Carvedilol competition binding in β_1 AR stable cells

a. co-IP of β_1 ARs and active $G\alpha_i$

b. ERK phosphorylation in HEK293 cells transfected with β AR chimera mutants.

Reviewers' comments:

Reviewer #1 (Remarks to the Author):

The authors have addressed my concerns by providing new data and explanations. The findings on the involvement of both Gai and b-arrestins in the carvedilol-induced b1AR signaling and EGFR-mediated internalization are compelling and novel. However, the direct molecular and functional link between Gai and b-arrestins still remain unresolved, in part because of the inability of the authors to demonstrate any complex formation between b1AR and b-arrestins (despite trying different approaches and reporting such interaction previously). The study has improved by the addition of new data, but I still have some concerns and issues, which are listed below.

The presentation and description of Fig. 5c are confusing, and inconsistent with what is presented in Fig. 5b. I suspect there is some kind of mistake with the labeling of the figure. Indeed, it is mentioned, “the depletion of Gai blocked carvedilol-induced EGFR internalization, whereas absence of Gai had no effect on EGF- or isoproterenol- induced responses”. However, Fig. 5C (far right bar chart) shows, that depletion of Gai had the same effect on both carvedilol- and isoproterenol-induced EGFR internalization ($p < 0.0001$ for both). I assume that what was meant to be shown here is the amount of receptors internalized as compared to NS condition, and not the amount of receptor left at the PM after agonist treatment? Either way, this is not consistent with what is shown in Fig. 5b. This needs to be shorted out.

On line 249, it is said that “Since GRK-mediated receptor phosphorylation is essential for b-arrestin recruitment to b1ARs, we used a mutant b1AR that lacks the putative GRK phosphorylation sites within the receptor carboxyl terminal tail (GRK252 b1AR) and therefore unable to recruit b-arrestins [9]”. I am puzzled here, since the authors are reporting that they are unable to show the interaction of b-arrestin with WT b1ARs upon carvedilol activation. How the authors can genuinely claim that GRK-mediated b1AR phosphorylation and b-arrestin recruitment are not required for carvedilol-induced Gai recruitment, after using a mutant b1AR lacking the putative GRK phosphorylation sites, since they are unable to detect b-arrestin binding to WT b1AR. In the same line, it could be argued that despite them not detecting b-arrestin binding to EGFR, this interaction occurs in cells.

The fact that carvedilol induces FRET changes in the b1AR conformation sensor is sensitive to b-arrestin removal is compelling for a role of b-arrestin in modulating the receptor. However, it doesn't prove that b-arrestin is recruited to the carvedilol-bound b1AR per se. Indeed, one can

agree that this effect is indirect, and occurs through third party interactions (such as EGFR or other membrane proteins), which could allosterically modulate β 1AR's conformation. Again, the demonstration that b-arrestin binding to β 1AR would clarify this point, and provide a mechanistic link between Gai and b-arrestin in the biased signaling of β 1AR. I appreciate the authors trying many approaches to detect such interaction, as well as being candid about their failure to do so, and for acknowledging not being able to reproduce such result. However, I also feel that the authors should be more forthright here by highlight these caveats in the discussion, and stating upfront that despite showing a role for both Gai and b-arrestin in the carvedilol-mediated biased signaling on β 1AR, the link between Gai and b-arrestin still remains unclear, and at least in the case of b-arrestin, this could be due to an indirect effect not occurring at the β 1AR level.

Reviewer #3 (Remarks to the Author):

As I previously stated in the initial review, this study on the β 1AR is timely and will be of interest to the GPCR field in general.

Numerous new data have been added to the initial manuscript and overall the authors have satisfactorily addressed most of the points raised in the initial review.

However, the persistent lack of demonstration that β -arrestins are recruited to the carvedilol-activated β 1AR, despite the efforts of the authors, is a substantial gap in the model they are proposing that requires more in depth discussion with alternative hypotheses (see below). It is indeed possible that, as a class A GPCR, the interaction of the β 1AR with β -arrestins is transient and of low affinity. However, the results of the multiple experiments using different approaches, suggest that there might not be a direct interaction between the receptor and β -arrestins. Furthermore, the additional FRET assay (Reviewer Fig. 2a) with the β 1AR conformation sensor in cells lacking β -arrestins indicate that they somehow participate (it may be indirect) to the stabilization of the carvedilol-bound β 1AR but does not demonstrate direct interaction of β -arrestins with the receptor.

It is stated in the discussion section, that "It is possible that carvedilol-induced β -arrestins-mediated β 1AR signaling does not involve a direct interaction between receptor and β -arrestins" (p.16). It is a rather short statement considering the above. The authors should be more open about that possibility, discuss it as a serious alternative hypothesis and along these lines propose alternative mechanisms.

Reviewers' comments:

Reviewer #1 (Remarks to the Author):

The authors have addressed my concerns by providing new data and explanations. The findings on the involvement of both Gai and b-arrestins in the carvedilol-induced b1AR signaling and EGFR-mediated internalization are compelling and novel. However, the direct molecular and functional link between Gai and b-arrestins still remain unresolved, in part because of the inability of the authors to demonstrate any complex formation between b1AR and b-arrestins (despite trying different approaches and reporting such interaction previously). The study has improved by the addition of new data, but I still have some concerns and issues, which are listed below.

The presentation and description of Fig. 5c are confusing, and inconsistent with what is presented in Fig. 5b. I suspect there is some kind of mistake with the labeling of the figure. Indeed, it is mentioned, "the depletion of Gai blocked carvedilol-induced EGFR internalization, whereas absence of Gai had no effect on EGF- or isoproterenol- induced responses". However, Fig. 5C (far right bar chart) shows, that depletion of Gai had the same effect on both carvedilol- and isoproterenol-induced EGFR internalization ($p < 0.0001$ for both). I assume that what was meant to be shown here is the amount of receptors internalized as compared to NS condition, and not the amount of receptor left at the PM after agonist treatment? Either way, this is not consistent with what is shown in Fig. 5b. This needs to be shorted out.

Thank you for pointing this out. We mistakenly switched the labeling of the EGF (E) and Carv (C) on the bar graph of Fig. 5c right panel. We have now corrected this in the revised manuscript, with the corrected E and C X-axis label highlighted in red. We also changed the color of the bars, making the bar color of each stimulation consistent with the color of the matching representative flow panel (Fig. 5c left panel). The data in the bar graph represents the remaining amount of EGFR on the plasma membrane, therefore the decrease represents the amount of EGFR internalized.

On line 249, it is said that "Since GRK-mediated receptor phosphorylation is essential for b-arrestin recruitment to b1ARs, we used a mutant b1AR that lacks the putative GRK phosphorylation sites within the receptor carboxyl terminal tail (GRK252 b1AR) and therefore unable to recruit b-arrestins [9]". I am puzzled here, since the authors are reporting that they are unable to show the interaction of b-arrestin with WT b1ARs upon carvedilol activation. How the authors can genuinely claim that GRK-mediated b1AR phosphorylation and b-arrestin recruitment are not required for carvedilol-induced Gai recruitment, after using a mutant b1AR lacking the putative GRK phosphorylation sites, since they are unable to detect b-arrestin binding to WT b1AR. In the same line, it could be argued that despite them not detecting b-arrestin binding to EGFR, this interaction occurs in cells.

We have previously shown that in response to the balanced agonists isoproterenol (Rapacciuolo et al., JBC, 2003) and dobutamine (Noma et al., JCI, 2007), GRK-mediated β 1AR phosphorylation is required for β -arrestin recruitment to the receptor. Additionally, Carvedilol has been shown to phosphorylate specific amino acid residues on the c-tail of the β 2AR (Nobles et al. Sci Sig, 2011). However, as this reviewer correctly points we have not demonstrated the same for carvedilol and the β 1AR and inferred from the above previous published data that this process of GRK-mediated phosphorylation of the c-tail to promote β -arrestin recruitment also

occurs following carvedilol stimulation. While it is clear that the mutant GRK- β 1AR cannot be phosphorylated on the c-tail since all serine and threonine residues have been mutated to alanine, we agree with the reviewer that we have overreached with our conclusion because we are not able to detect a direct β 1AR- β -arrestin interaction in response to carvedilol stimulation. To address this concern, we have substantively modified the text softening our conclusion. The changes to the text can be found on lines 248-281 of page 11-12 of the revised manuscript and also copied below.

“GRK-mediated receptor phosphorylation plays a critical role in β -arrestin-dependent signaling of β ARs³². When β 1ARs are stimulated by balanced agonists, such as isoproterenol or dobutamine, GRK-mediated β 1AR phosphorylation of the carboxyl-terminal tail occurs and is required for agonist mediated β -arrestin recruitment^{9,33}. For the β 2AR, a similar process has been shown to occur whereby stimulation with the β -arrestin-biased agonist carvedilol promotes β 2AR phosphorylation at specific GRK sensitive amino acid residues 355 and 356 of the c-terminal tail^{21,32}. Here, we sought to determine whether GRK-mediated phosphorylation of the β 1AR is a necessary step in the carvedilol-induced $G\alpha_i$ recruitment to the receptor. To address this question, we used a mutant β 1AR that lacks the putative GRK phosphorylation sites within the receptor carboxyl-terminal tail (GRK- β 1AR) and therefore unable to be phosphorylated by GRKs^{9,33}. We show that carvedilol stimulation increased $G\alpha_i$ recruitment to a similar extent between wild type and GRK- β 1ARs, as assessed by co-immunoprecipitation (Fig. 7a) and suggests that GRK-mediated β 1AR phosphorylation is not required for carvedilol-induced $G\alpha_i$ recruitment to the β 1AR.”

The fact that carvedilol induces FRET changes in the b1AR conformation sensor is sensitive to b-arrestin removal is compelling for a role of b-arrestin in modulating the receptor. However, it doesn't prove that b-arrestin is recruited to the carvedilol-bound b1AR per se. Indeed, one can argue that this effect is indirect, and occurs through third party interactions (such as EGFR or other membrane proteins), which could allosterically modulate b1AR's conformation. Again, the demonstration that b-arrestin binding to b1AR would clarify this point, and provide a mechanistic link between $G\alpha_i$ and b-arrestin in the biased signaling of b1AR. I appreciate the authors trying many approaches to detect such interaction, as well as being candid about their failure to do so, and for acknowledging not being able to reproduce such result. However, I also feel that the authors should be more forthright here by highlight these caveats in the discussion, and stating upfront that despite showing a role for both $G\alpha_i$ and b-arrestin in the carvedilol-mediated biased signaling on b1AR, the link between $G\alpha_i$ and b-arrestin still remains unclear, and at least in the case of b-arrestin, this could be due to an indirect effect not occurring at the b1AR level.

We agree with the reviewer that based on our data we cannot rule out the possibility of an indirect role of β -arrestin until techniques with higher resolution become available. To address this concern, we have more thoughtfully discussed the possible mechanism for β -arrestin involvement in carvedilol-stimulated β 1AR signaling, and removed claims related to the role of $G\alpha_i$ in β -arrestin recruitment to the receptor. The changes to the text can be found on lines 374-404 on page 16-17 of the revised manuscript and also copied below.

“While our data show that both $G\alpha_i$ and β -arrestins are required for carvedilol-induced biased signaling of the β 1AR, whether β -arrestin is recruited to the carvedilol occupied β 1AR

remains to be determined. Using a number of methodologies, such as co-immunoprecipitation, confocal- or bioluminescence resonance energy transfer-based assays, we were unable to detect carvedilol-induced β -arrestin recruitment to the β_1 AR. This may be due to a number of reasons: 1) ligand-induced β -arrestin recruitment and activation is rapid, within 2 second after stimulation, and reversible⁴¹; 2) the affinity of the β_1 AR- β -arrestin interaction is low. Both the β_1 AR and the β_2 AR are known as class A receptors, since they are characterized by transient and weak interaction with β -arrestins along with a rapid recycling to the plasma membrane after internalization. To demonstrate carvedilol triggered β -arrestin recruitment to the β_2 AR, previous studies used a chimeric receptor consisting of the β_2 AR fused to vasopressin V₂ receptor cytoplasmic tail (β_2 AR-V₂R) to increase the affinity of β -arrestin to the ligand occupied receptor²¹. However, as we have shown (Fig. 7c), the C-tail of the β_1 AR is required for G α_i recruitment. Therefore substituting the β_1 AR C-tail with the V₂R tail would not provide a chimeric receptor suitable to study the role of G α_i in carvedilol stimulated β -arrestin recruitment. Importantly, we cannot exclude that the carvedilol-stimulated β_1 AR signaling is mediated by β -arrestin by an indirect mechanism that does not require direct binding of β -arrestin to the β_1 AR. A recent study identified unique features for the β_1 AR with respect to β -arrestin interaction and activation⁴², where a brief interaction with the activated β_1 AR is sufficient to target β -arrestin2 to clathrin-coated structures and trigger ERK signaling even in the absence of receptor association⁴². This β -arrestin “activation at a distance” mechanism suggests that a β_1 AR- β -arrestin complex may not be essential for the activation of β -arrestin-dependent signaling and could explain our findings for a role of β -arrestin in carvedilol-induced signaling without a direct β_1 AR- β -arrestin interaction. Lastly, it is also possible that instead of directly engagement with the β_1 AR, β -arrestins could associate with other components of the signaling cascade such as the transactivated EGFR. This has recently been shown for the V2 vasopressin receptor signaling, where β -arrestins are recruited to, and act downstream, of the transactivated insulin-like growth factor receptor⁴³.”

Reviewer #3 (Remarks to the Author):

As I previously stated in the initial review, this study on the β_1 AR is timely and will be of interest to the GPCR field in general.

Numerous new data have been added to the initial manuscript and overall the authors have satisfactorily addressed most of the points raised in the initial review.

However, the persistent lack of demonstration that β -arrestins are recruited to the carvedilol-activated β_1 AR, despite the efforts of the authors, is a substantial gap in the model they are proposing that requires more in depth discussion with alternative hypotheses (see below). It is indeed possible that, as a class A GPCR, the interaction of the β_1 AR with β -arrestins is transient and of low affinity. However, the results of the multiple experiments using different approaches, suggest that there might not be a direct interaction between the receptor and β -arrestins. Furthermore, the additional FRET assay (Reviewer Fig. 2a) with the β_1 AR conformation sensor in cells lacking β -arrestins indicate that they somehow participate (it may be indirect) to the stabilization of the carvedilol-bound β_1 AR but does not demonstrate direct interaction of β -arrestins with the receptor.

It is stated in the discussion section, that “It is possible that carvedilol-induced β -arrestins-mediated β_1 AR signaling does not involve a direct interaction between receptor and β -arrestins” (p.16). It is a rather short statement considering the above. The authors should be more open

about that possibility, discuss it as a serious alternative hypothesis and along these lines propose alternative mechanisms.

This concern is the same as Reviewer #1 and our response is copied below.

We agree with the reviewer that based on our data we cannot rule out the possibility of an indirect role of β -arrestin until techniques with higher resolution become available. To address this concern, we have more thoughtfully discussed the possible mechanism for β -arrestin involvement in carvedilol-stimulated β_1 AR signaling, and removed claims related to the role of G_{α_i} in β -arrestin recruitment to the receptor. The changes to the text can be found on lines 374-404 on page 16-17 of the revised manuscript and also copied below.

“While our data show that both G_{α_i} and β -arrestins are required for carvedilol-induced biased signaling of the β_1 AR, whether β -arrestin is recruited to the carvedilol occupied β_1 AR remains to be determined. Using a number of methodologies, such as co-immunoprecipitation, confocal- or bioluminescence resonance energy transfer-based assays, we were unable to detect carvedilol-induced β -arrestin recruitment to the β_1 AR. This may be due to a number of reasons: 1) ligand-induced β -arrestin recruitment and activation is rapid, within 2 second after stimulation, and reversible⁴¹; 2) the affinity of the β_1 AR- β -arrestin interaction is low. Both the β_1 AR and the β_2 AR are known as class A receptors, since they are characterized by transient and weak interaction with β -arrestins along with a rapid recycling to the plasma membrane after internalization. To demonstrate carvedilol triggered β -arrestin recruitment to the β_2 AR, previous studies used a chimeric receptor consisting of the β_2 AR fused to vasopressin V_2 receptor cytoplasmic tail (β_2 AR- V_2 R) to increase the affinity of β -arrestin to the ligand occupied receptor²¹. However, as we have shown (Fig. 7c), the C-tail of the β_1 AR is required for G_{α_i} recruitment. Therefore substituting the β_1 AR C-tail with the V_2 R tail would not provide a chimeric receptor suitable to study the role of G_{α_i} in carvedilol stimulated β -arrestin recruitment. Importantly, we cannot exclude that the carvedilol-stimulated β_1 AR signaling is mediated by β -arrestin by an indirect mechanism that does not require direct binding of β -arrestin to the β_1 AR. A recent study identified unique features for the β_1 AR with respect to β -arrestin interaction and activation⁴², where a brief interaction with the activated β_1 AR is sufficient to target β -arrestin2 to clathrin-coated structures and trigger ERK signaling even in the absence of receptor association⁴². This β -arrestin “activation at a distance” mechanism suggests that a β_1 AR- β -arrestin complex may not be essential for the activation of β -arrestin-dependent signaling and could explain our findings for a role of β -arrestin in carvedilol-induced signaling without a direct β_1 AR- β -arrestin interaction. Lastly, it is also possible that instead of directly engagement with the β_1 AR, β -arrestins could associate with other components of the signaling cascade such as the transactivated EGFR. This has recently been shown for the V_2 vasopressin receptor signaling, where β -arrestins are recruited to, and act downstream, of the transactivated insulin-like growth factor receptor⁴³.”